# Seismic Hazard in Greece: A Comparative Study for the Region of East Macedonia and Thrace

Dimitris Sotiriadis [1,*], Basil Margaris [2], Nikolaos Klimis [1] and Ioannis M. Dokas [1]

1   Department of Civil Engineering, Democritus University of Thrace, University Campus, 67100 Xanthi, Greece; idokas@civil.duth.gr (I.M.D.)
2   Institute of Engineering Seismology and Earthquake Engineering, EPPO-ITSAK, 55535 Thessaloniki, Greece
*   Correspondence: dsotiria@civil.duth.gr

**Abstract:** Greece is located in one of the most seismically active regions in Europe. Many seismic hazard studies have been performed for various sites around Greece, at a regional or local scale. However, the latest national seismic hazard map, currently used for the seismic design of buildings and infrastructure, was published in 2000 and has not been updated since then. In light of recent advances in seismic source and ground motion modeling, the present study focuses on a comparative Probabilistic Seismic Hazard Assessment (PSHA) for the region of East Macedonia and Thrace (EMTH), located in Northern Greece. Various seismic source models are implemented and compared against an updated earthquake catalog to form the necessary source model logic tree. The ground motion logic tree is composed of Ground Motion Prediction Equations (GMPEs), which have been proven suitable for implementation in Greece. PSHA results are presented for the most important cities of East Macedonia and Thrace in a comparative way, which highlights the variability of the seismic hazard among the various seismic source models. An updated seismic hazard map of the study area is proposed, and a comparative disaggregation analysis is performed to estimate the earthquake scenarios with the largest contribution to the seismic hazard.

**Keywords:** seismic hazard assessment; ground motion; seismic disaggregation; earthquake scenarios





## 1. Introduction

Greece is located in one of the most seismically active regions in Europe. Many seismic hazard studies have been performed for various places around Greece, at a regional or local scale [1–12]. However, the latest national seismic hazard map, currently used for the seismic design of buildings and infrastructure, was published in 2000, revised in 2003 [13], and has not been updated since then.

In light of recent advances in seismic source and ground motion modeling and an ongoing project regarding the Climate Crisis and Civil Protection in Greece (Risk and Resilience Assessment Center, riskac.eu), the aim of the present study is the assessment of seismic hazards for the Region of East Macedonia and Thrace (REMTH), located in the north-eastern part of Greece. The study area borders Bulgaria in the north, Turkey in the east, and the Northern Aegean Sea to the south, as shown in Figure 1. The most important seismotectonic/faulting feature of the broader region is close to the edge of the study area and consists of the branch of the North Anatolian fault, a strike-slip fault that ends up in the Northern Aegean Sea. Figure 2 presents the main seismic faults of REMTH, as presented in [14], along with historical and modern seismicity data, according to [15]. Within the mainland of REMTH, large normal faults, striking almost in the WE direction, have been identified. The region has been affected by earthquake events that have occurred outside its administrative borders and are associated with mainly normal faults striking from WE to NW-SE, as shown in Figure 2.

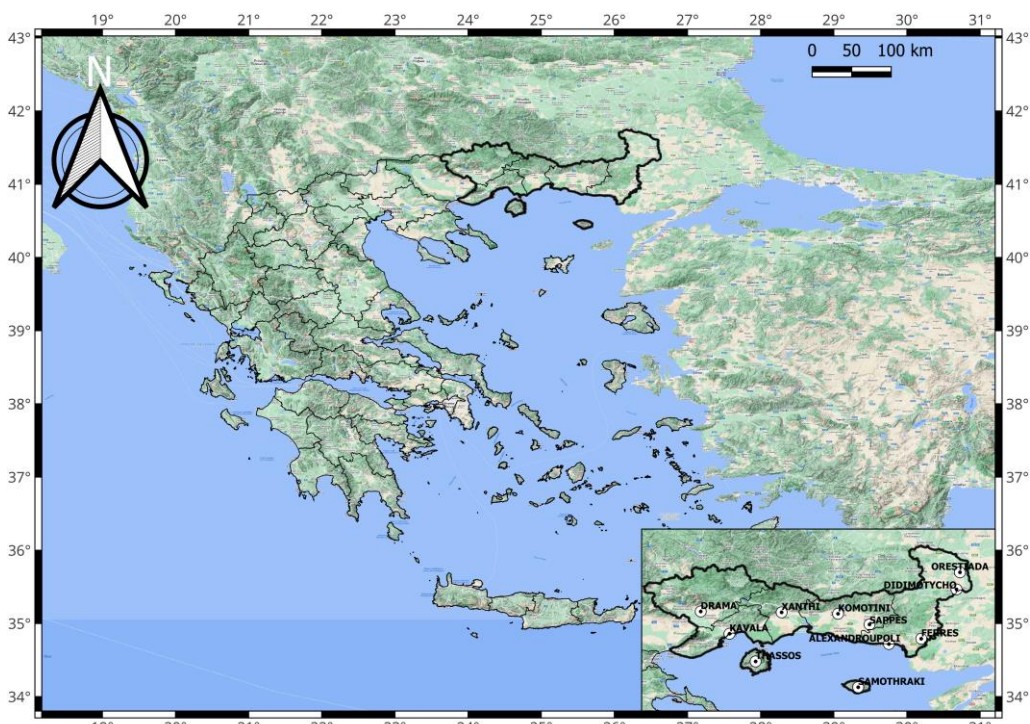

**Figure 1.** Location of the study area. The embedded map shows a more detailed view of the study area (REMTH), along with the most important cities and towns.

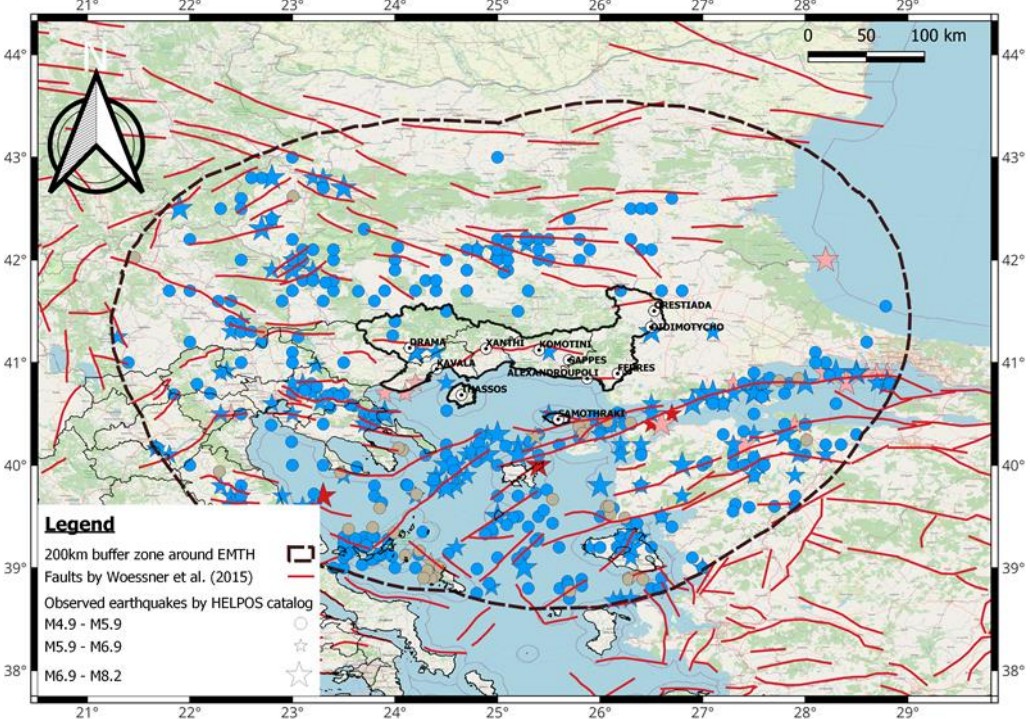

**Figure 2.** Main seismic faults and seismicity of the broader region of REMTH [14]. The seismicity covers the period between 479 B.C.E. and 2018 C.E [15]. The red earthquake events occurred between 479 B.C.E. and 0 C.E., the orange-colored events occurred between 0 and 1000, the blue-colored events occurred between 1000 and 2000, and the gray ones between 2000 and 2018.

Figure 2 presents the location of the epicenters of significant earthquakes (with magnitudes M ≥ 4.9) that have affected the study area. As Figure 2 denotes, within the borders

of the EMTH mainland, a few significant earthquakes with magnitudes M5.9–M6.9, as well as some smaller magnitude events, have been reported between 1000 and 2000. However, significant seismic activity has been observed near the borders of REMTH. More specifically, close to the south edge of REMTH, large-magnitude earthquakes have been reported historically (479 B.C.E.–0 C.E.), which were associated with the seismic activity of the North Anatolian fault. Several strong seismic events have affected the study area significantly [16]. In 597, an earthquake of magnitude M6.8 occurred close to Kavala, which destroyed the town of Filippoi and changed the flow of the Strymon River. On the 29 July 1752, an earthquake of magnitude M7.5 occurred in eastern Thrace, caused significant damage to structures in Edirne, Turkey, and was strongly felt in several sites within REMTH, with macroseismic intensities ($I_{MM}$) ranging between 6 and 9. On 6 November 1784, an earthquake of magnitude M6.7 struck the city of Komotini. According to the monks' notes, 500 houses were completely destroyed. On 5 May 1829, an earthquake of magnitude M7.3 struck the city of Drama, whereas a foreshock occurred on 13 April of the same year. The mainshock completely destroyed the cities of Drama ($I_{MM}$ = 10) and Xanthi ($I_{MM}$ = 9), and severe damage was reported in Eleftheroupoli and Kavala ($I_{MM}$ = 8). On the 6 August 1860, an M6.2 event occurred offshore, close to the island of Samothraki, which caused moderate damage to buildings, ground cracking, and rockfalls. On the 14 June 1864, an M7.0 earthquake hit the town of Genisea, close to Xanthi, with several houses collapsing in Genisea and Porto Lagos. The city of Drama was hit by an M6.0 event on 29 March 1867, which caused the collapse of most of the houses in the city. Samothraki was hit by an M6.8 event in February 1893. Out of 600 houses on the island, 52 collapsed, whereas 300–350 suffered extensive damage. The maximum observed intensity was $I_{MM}$ 9 on the island, whereas intensities between 5 and 7 were observed in other sites of REMTH. On the 18 April 1928, an earthquake of magnitude M7.0 occurred in Bulgaria, with a foreshock of M6.8 occurring four days earlier. Within REMTH, the $I_{MM}$ ranged between 4 and 6, with the city of Komotini being the one that was affected the most. Other events that have affected regions of REMTH were the M7.0 event that struck close to Ierissos (1928) and the M7.0 earthquake of Kallipoli, Turkey (1975), whereas the most recent event is dated 9 November 1985, which hit the city of Drama and had a magnitude of M5.5 and a maximum $I_{MM}$ equal to 7.

Herein, to assess the seismic hazard in REMTH, the approach of Probabilistic Seismic Hazard Assessment (PSHA) [17] is implemented along with the logic tree approach to take into account the uncertainty in both seismic source and ground motion modeling. The selection of the Ground Motion Prediction Equations (GMPEs) is based on the work of Sotiriadis and Margaris (2023) [18], who have evaluated many global, European, and local GMPEs through data-driven methods based on the comparison between their estimates and strong-motion data from Greece. Moreover, various seismic source models that have been proposed during the last few years for Greece are considered in this study. The seismicity posed by these models for the study area is compared against seismicity data from a recently published earthquake catalog for Greece [15]. PSHA is performed, including many ground motion intensity measures and return periods. Emphasis is given to the variability of the estimated seismic hazard of the study area due to the different seismic source models that are used. An updated PSHA map is proposed for REMTH and compared with earlier regional and international maps. Subsequently, the disaggregation of the seismic hazard is performed for the most significant cities of REMTH for all the considered seismic source models separately, and the seismic scenarios that contribute the most are identified. The work presented herein highlights the seismic hazard in a region that has not been studied thoroughly in the past while pointing out the variations in hazard estimates due to the seismic source model that is adopted.

## 2. Materials and Methods

### 2.1. Seismic Source Models

Distributed seismicity source models are the most common for implementation in PSHA studies at a national or regional scale. They consist of area sources with uniform seismotectonic features within their borders. In each source, the seismicity parameters are defined based on earthquake catalogs of recorded and historical seismic events. Such seismicity models have been published for Greece by many researchers [19–22]. Furthermore, distributed seismicity models have been developed for wide geographical regions [14,23] or for site-specific, large projects of great extent and importance [11], which include the Greek territory or a part of it. Besides the geometry of the area sources, their basic seismotectonic features include the earthquake magnitude–frequency (M–F) distribution and the maximum earthquake magnitude ($M_{max}$) that the seismic source is able to generate. The M–F distribution is usually defined based on the observed seismicity within an area source and is usually modeled through a truncated Gutenberg–Richter distribution.

Another modeling approach to the seismicity of a region includes the seismically active faults (fault-based seismicity), which are associated with large magnitude earthquakes (e.g., $M \geq 6.0$ or 6.5), accompanied by background seismicity area sources, which are associated with lower magnitude events. The characteristics of seismic faults, which have caused known strong earthquakes in Greece and the surrounding regions, have been identified in [24]. The estimation of their M–F distribution and $M_{max}$ was based on the recorded seismicity as well as on the fault geometry through empirical relationships that related the rupture geometry to the earthquake magnitude [25–28]. Moreover, European seismic hazard models [14,23], as well as specific project studies [11], have proposed seismotectonic features for seismic faults in Greece, the geometry of which is similar to the ones proposed by [29]. The rate of earthquake occurrence on those faults has been defined through geodetic and geological methods, which give estimates of their geological slip rate. Subsequently, utilizing the estimated slip rate and the $M_{max}$, which is defined in a similar procedure to the area sources, well-known methodologies are implemented [30] to obtain the M–F distribution.

Within the context of the present study, seismic source models that are published in the literature were utilized. The intention of the authors was to implement both distributed and fault-based seismicity source models, whose seismotectonic features have been defined based on recorded seismicity and geological slip rates. The seismic source models (SM) that were implemented are described in Table 1 and presented in Figure 3. In Figure 3, a buffer zone of 200 km around REMTH is shown, which denotes the maximum distance considered in the subsequent PSHA analyses. The TAP_faults_SHAREB source model's geometry basically coincides with the one of SHARE_FB (Figure 3b) [11], hence it is not repeated in the figure. The same background seismicity area sources were considered for the fault-based seismicity models. They were originally implemented along with the faults of [14] for earthquake events M < 6.5. However, for the implementation of the PZ01_SHAREFB source model, modifications were made so that they apply for earthquake events with M < 6.0 since the faults indicated by [24] are associated with earthquakes with $M \geq 6.0$. Moreover, it should be noted that the sources identified by the fault-based source models are not actual faults in the geologic sense but Composite Seismogenic Sources (CSS) that may include one or more fault segments [31].

**Table 1.** Description of the Seismic source models considered in the present study.

| SM Designation | Description |
| --- | --- |
| PZ01_SHAREB | Seismic faults by [24] and background seismicity area sources by [14] |
| SHARE_FB | Seismic faults and background seismicity area sources by [14] |
| TAP faults_SHAREB | Seismic faults by [11] and background seismicity area sources by [14] |
| SHARE_Area | Area sources of distributed seismicity by [14] |

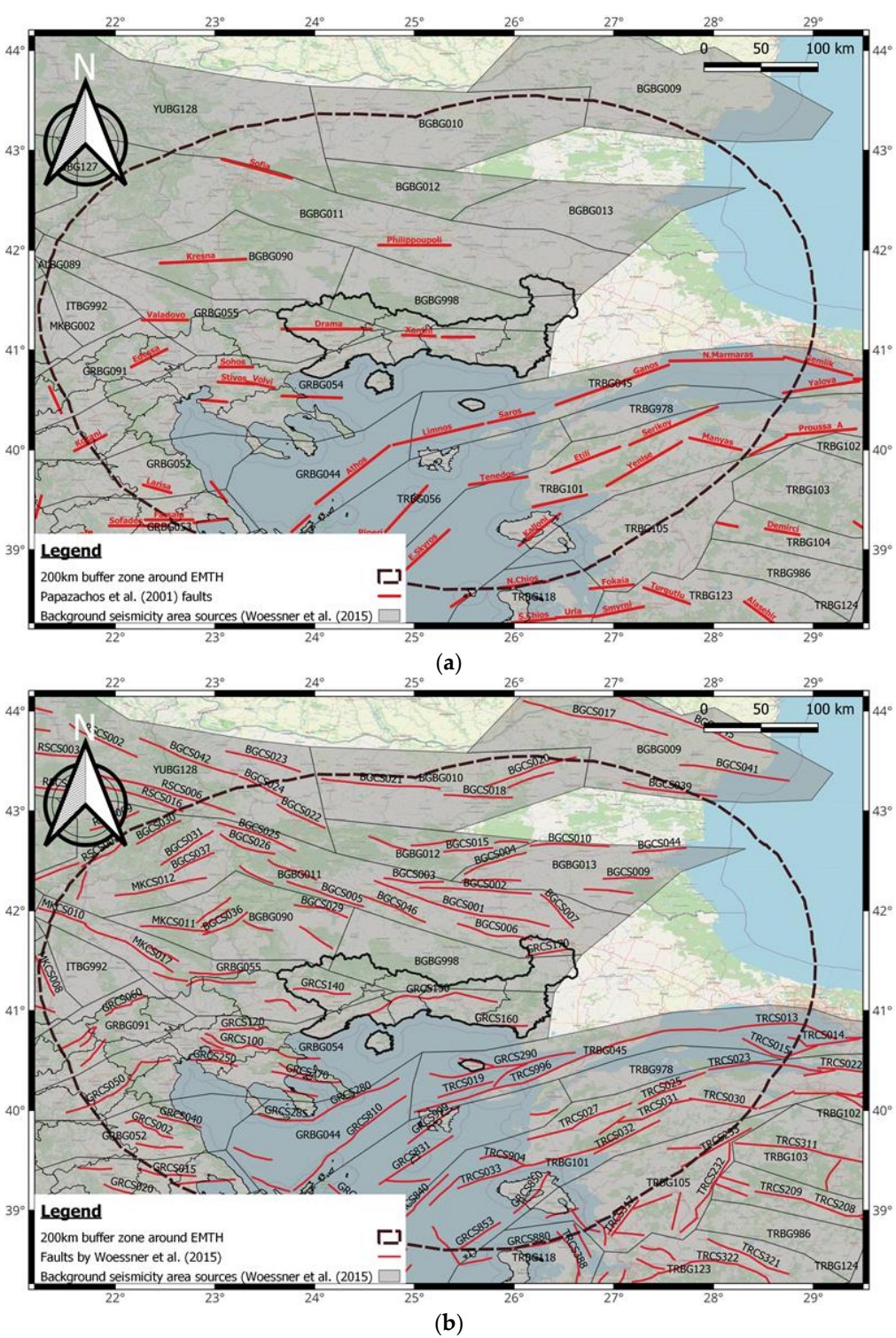

**Figure 3.** *Cont.*

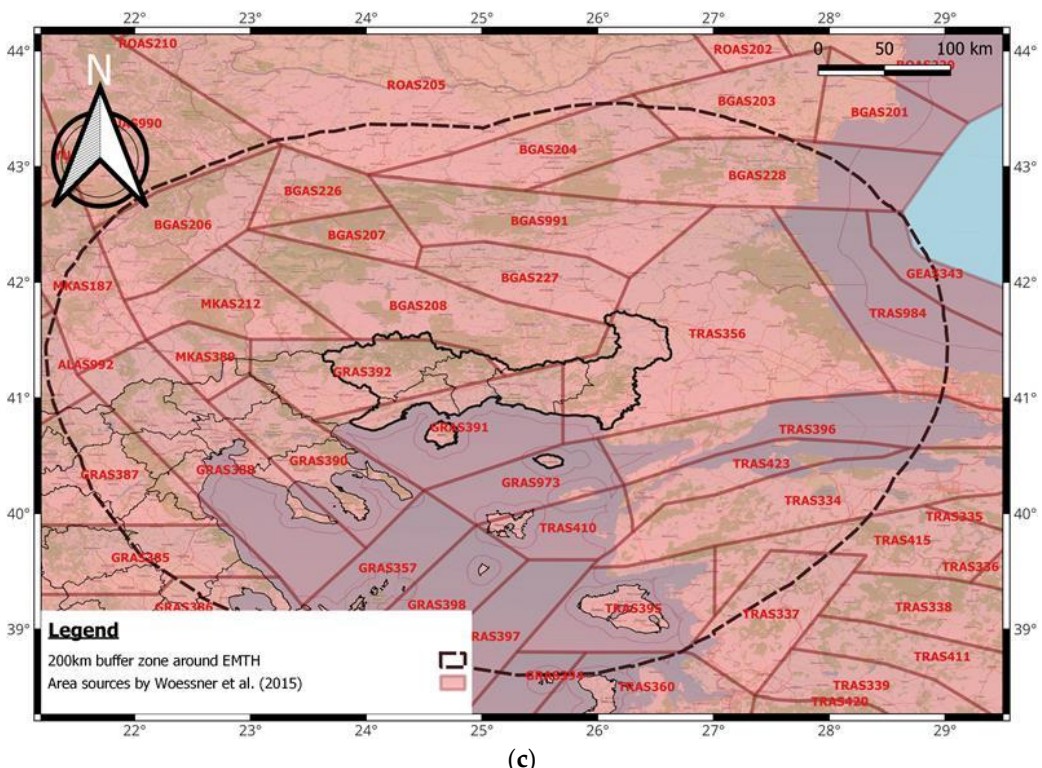

**(c)**

**Figure 3.** Investigated seismic source models in the present study. (**a**) PZ01_SHAREB [14,24], (**b**) SHARE_FB [14], and (**c**) SHARE_Area [14].

Each of the considered SMs comes with its own M–F distribution, which is described by a Gutenberg–Richter distribution (GR) and describes the seismicity of each source. To obtain an overview of the seismicity rate forecast of the considered SMs, the total expected average annual earthquake rates were calculated and compared with the cumulative M–F distribution of the earthquake catalog of [15] in Figure 4. For all the considered SMs, this calculation involved summing up the rates of all individual zones [14].

Compared with the observed seismicity, the SHARE_FB SM highly overestimates the observed earthquakes with a magnitude lower than M6.5, whereas it fits well with the recorded seismicity for larger magnitudes (M > 6). It should be noted that for this SM, the seismic events with a magnitude lower than M6.5 are controlled by the background seismicity area sources. The mean residual between the observed annual number of events over the whole magnitude range (in log terms) is −0.17. The SHARE_Area SM seems to fit reasonably well with the observed seismicity over the whole range of observed earthquake magnitudes, although a slight overestimation is apparent for magnitudes between M5.3 and M5.9 and a slight underestimation for M6.3 and M6.7. The mean residual between the observed annual number of events over the whole magnitude range (in log terms) is −0.03. The PZ01_SHAREB SM also overestimates the seismic events with a magnitude up to M6.0, although not as much as the SHARE_FB SM. Moreover, it seems to underpredict the occurrence of earthquake events with a magnitude larger than M6.5. The mean residual between the observed annual number of events over the whole magnitude range (in log terms) is −0.09. The trends of the fit of the TAP faults_SHAREB seismicity to the observed seismicity are similar to the PZ01_SHAREB, while the mean residual between the observed annual number of events over the whole magnitude range (in log terms) is −0.06. It should be noted that the background seismicity model of the fault-based SMs is the same, with the difference between SHARE_FB, PZ01_SHAREB, and TAP faults_SHAREB being that for the latter, the background seismicity controls events with a magnitude lower than M6.0 rather than M6.5. Therefore, this modification leads to smaller discrepancies between the seismicity forecast of the SMs and the observed seismicity and lower mean residuals.

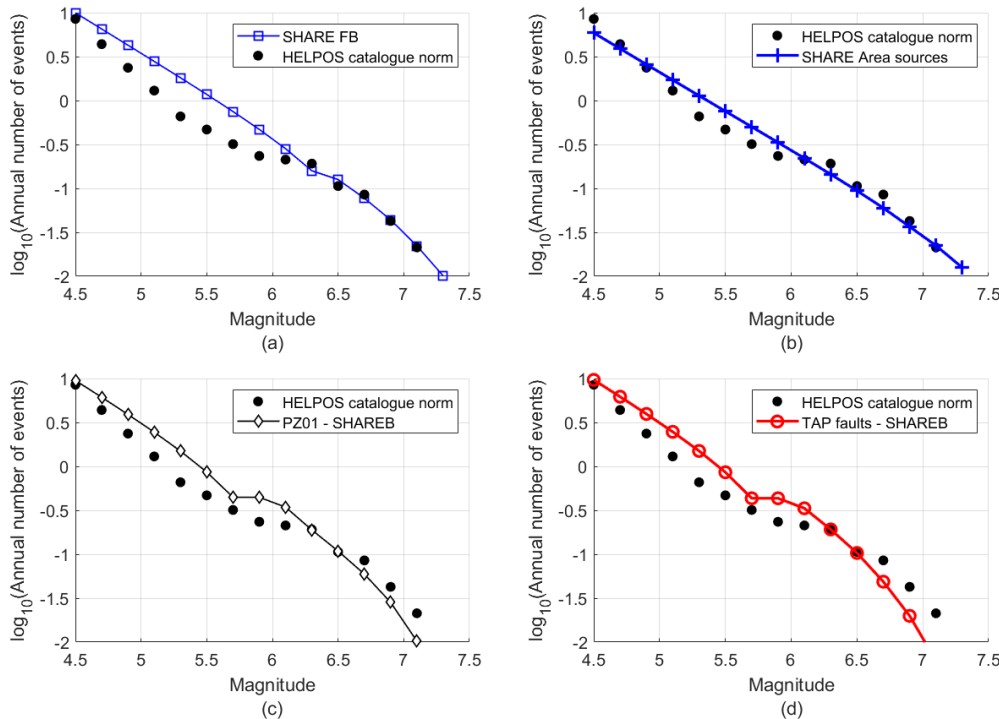

**Figure 4.** Total annual earthquake rate for crustal earthquakes for the study area, as denoted by the HELPOS earthquake catalog (black dots) and the investigated seismic source models (**a**) SHARE_FB, (**b**) SHARE_Area, (**c**) PZ01_SHAREB, and (**d**) TAP faults_SHAREB.

## 2.2. Ground Motion Modeling

One of the main components of PSHA is ground motion modeling. Quite recently, an updated GMPE has been proposed for Greece by [32], which has been calibrated based on the most updated strong motion database for Greece [33]. Although the GMPE of Boore et al. (2021) is currently the most reliable tool to estimate the ground motion in Greece, within the context of PSHA, more GMPEs should be implemented to construct a ground motion logic tree so that the associated epistemic uncertainty is reduced. Recently, [18] evaluated and ranked several GMPEs using the statistical methods of Log-likelihood (LLH) [34], Multivariate LLH [35], and Euclidean Distance Ranking (EDR) [36] and the most updated strong motion dataset for Greece. Their study included multiple ground motion intensity measures, such as the Peak Ground Acceleration (PGA) and Velocity (PGV), as well as some 5%-damped spectral acceleration ordinates ($S_a$). Using an unnormalized weighting scheme, they concluded with a final ranking and proposed the use of the top three GMPEs along with their associated weights. Their suggestion was adopted herein, and the selected GMPEs and their weights are shown in Table 2.

**Table 2.** Groun Motion Prediction Equations used in the present study.

| Ranking | GMPE | Weight |
|---------|------|--------|
| 1 | Boore et al. (2021) [32] | 0.38 |
| 2 | Kotha et al. (2020; 2022) [37,38] | 0.34 |
| 3 | Chiou and Youngs (2014) [39] | 0.28 |

## 2.3. Probabilistic Seismic Hazard Assessment (PSHA)

The seismic hazard at a specific site is expressed in terms of annual exceedance rates $\lambda_y$ or ground motion intensity measures (e.g., peak ground acceleration, PGA). For example, $\lambda_y$ is the number of earthquakes per year for which PGA is exceeded at a site. The inverse of $\lambda_y$ is equal to the mean return period (TR) in years. A basic assumption of PSHA is that

earthquake occurrence follows a Poisson process. Therefore, the probability that $y$ will exceed a specified value $Y$ within a given investigation time $t$ is calculated according to Equation (1).

$$P(y > Y) = 1 - e^{-t \cdot \lambda_y(Y)}$$ (1)

Using Equation (1) for various levels of $y$, the seismic hazard curves can be constructed, which denote the probability of exceeding the specified x values during a time period, $t$. In PSHA, $\lambda_y$ is computed through the total probability theorem [17], which is expressed through Equation (2).

$$\lambda_y = \sum_1^{N_s} v_i \int_{M_0}^{M_{max}} \int_{R_{min}}^{R_{max}} P[Y > y | M = m, R = r] f_M(m) f_R(r) dm dr$$ (2)

In Equation (2), $v_i$ is the mean rate of occurrence of earthquakes with a magnitude of $M_0$ for the seismic source $i$, $f_M$ $(m)$ and $f_R$ $(r)$ are the probability density functions of magnitude and distance, respectively, and $N_S$ is the number of seismic sources that are taken into account.

For the present study, the lowest earthquake magnitude, $M_0$, which was considered for all the seismic sources, was M4.5. Moreover, the investigation time, t, was set at 50 years. Furthermore, the truncation level of the GMPEs was set to 2σ, where σ is the standard deviation of a GMPE, so that unrealistically high values of ground motion intensity are avoided for large return periods. The PSHA calculations were performed in the Openquake Engine [40] for rock site conditions. Due to the implementation of GMPEs, which have been proven appropriate in Greece, the uncertainty in strong motion prediction for the present study could be considered more constrained compared with the uncertainty associated with seismic source modeling. To assess the PSHA results variability due to the seismic source models adopted, four PSHA logic trees were constructed. Each logic tree consists of one source model (Table 1) and three GMPEs with weights defined in Section 2.3 and Table 2. The intensity measures of ground motion considered were the peak ground acceleration and velocity (PGA and PGV), as well as spectral accelerations at various period values between 0.05 s and 4.0 s.

## 3. Results

### 3.1. PSHA Results

The seismic hazard curves of PGA and PGV are presented in Figures 5 and 6 for eleven significant sites in REMTH. In these curves (Figures 5 and 6), the PGA and PGV values are plotted with respect to the corresponding return period. The seismic hazard curves calculated through the Openquake Engine describe the probability of exceeding specific PGA and PGV values $P(y > Y)$, as shown in Equation (1). Utilizing Equation (1) and the fact that the inverse of the annual rate of exceedance, $\lambda_y$, is equal to the return period, the return period is calculated according to Equation (3).

$$TR = -t / \log(1 - P(y > Y))$$ (3)

The seismic hazard curves indicate that the intensity of ground motion varies within the entire region. There are sites located in the eastern part of REMTH, like Orestiada, Didymotycho, and Ferres, with low-intensity ground motion. On the other hand, the islands of Samothraki, Komotini, Drama, and Xanthi are characterized by relatively high values of PGA and PGV. A general observation coming from Figures 5 and 6 is that there is no source model that constantly provides the highest or lowest intensity of ground motion. For Kavala, Thasos, Sappes, Didymotycho, and Orestiada, the SHARE_Area source model clearly leads to the highest PGA and PGV values. However, for Drama and Ferres, the PSHA estimates of the SHARE_Area SM are close to the ones provided by the PZ01_SHAREB and TAP faults_SHAREB, respectively. Moreover, for some sites like

Drama, Xanthi, Komotini, and Sappes, the SHARE_FB source model leads to the lowest PGA and PGV values over the entire range of return periods.

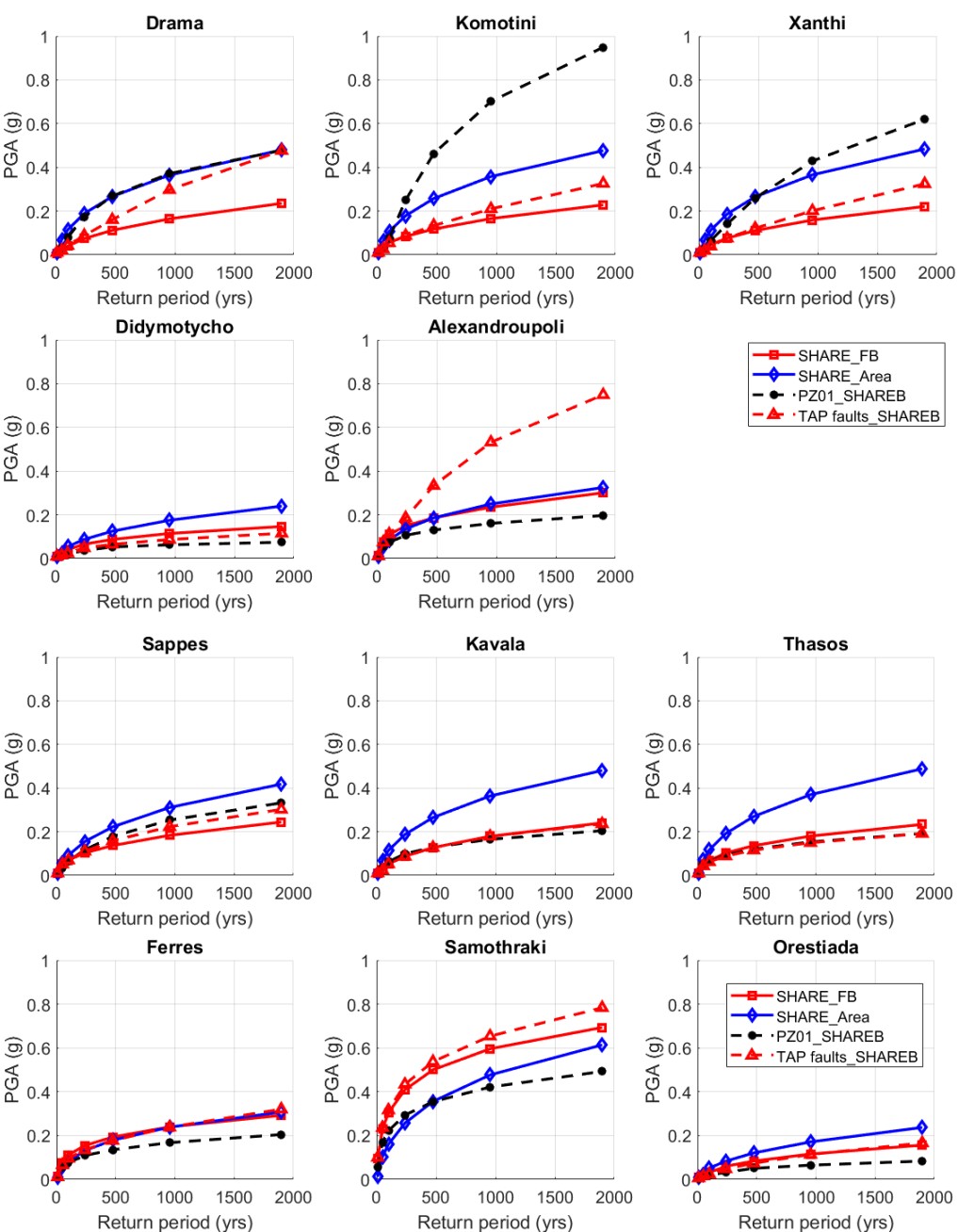

**Figure 5.** PSHA results for eleven important sites of REMTH, namely Drama, Komotini, Xanthi, Didymotycho, Alexandroupoli, Sappes, Kavala, Thasos, Ferres, Samothraki and Orestiada, in terms of PGA seismic hazard curves.

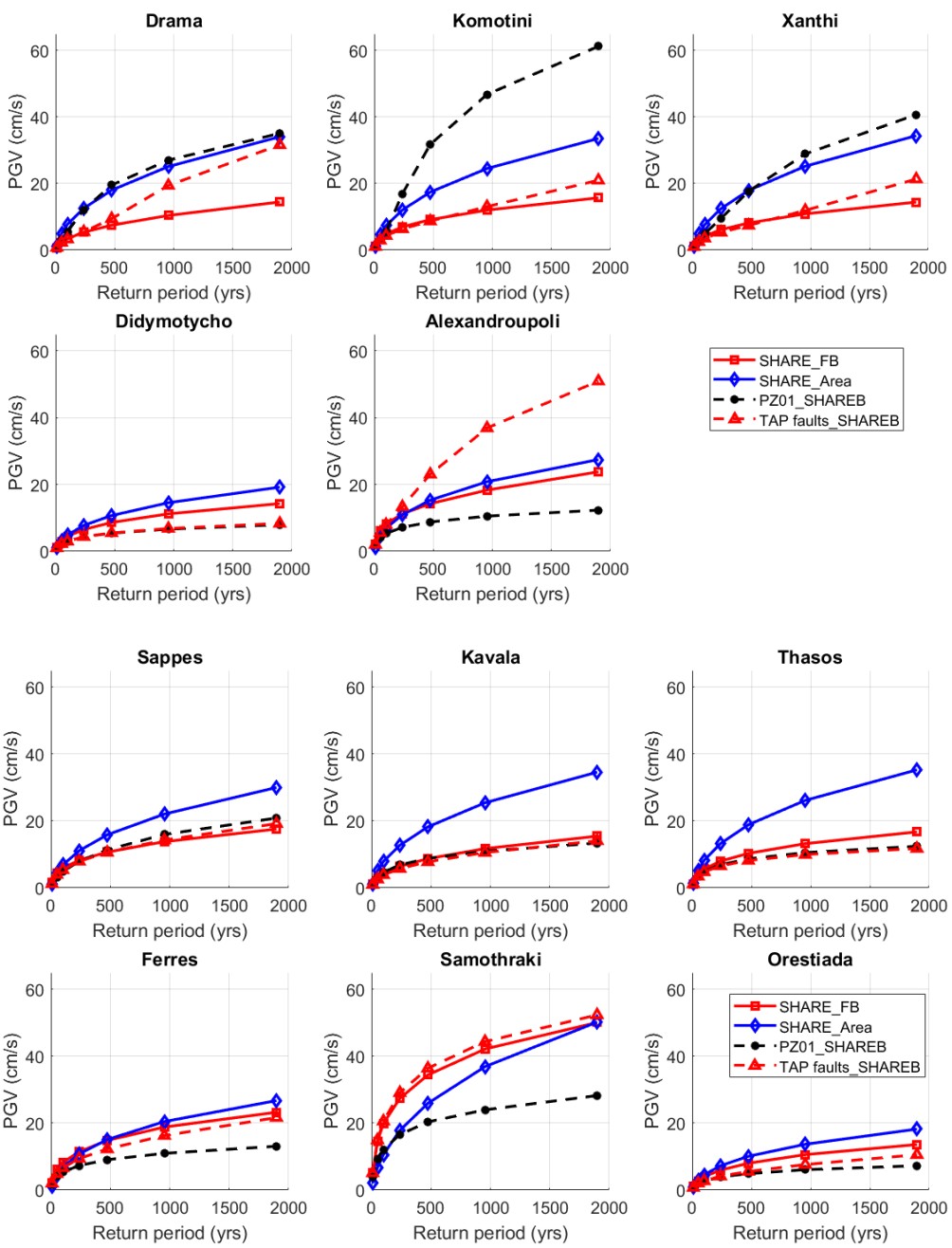

**Figure 6.** PSHA results for eleven important sites of REMTH, namely Drama, Komotini, Xanthi, Didymotycho, Alexandroupoli, Sappes, Kavala, Thasos, Ferres, Samothraki and Orestiada, in terms of PGV seismic hazard curves.

The variability of the intensity of ground motion due to the source model is depicted in Figure 7, where the standard deviation of PGA and PGV values, considering all the source models, is plotted against the return period for the important cities and towns of REMTH. For all sites, the standard deviation of ground motion increases with increasing return periods. However, three groups of sites can be identified in Figure 7. The red curves correspond to sites with a large standard deviation, especially for return periods larger than 250 years. These curves correspond to Komotini and Alexandroupoli. Both sites exhibit their largest ground motion intensity values when fault-based source models are considered, namely the PZ01_SHAREB and TAP faults_SHAREB, respectively. The standard deviation of PGA and PGV for these sites initiates at small values for low return periods and increases rapidly as the return period increases. The blue curves correspond to

sites with a low standard deviation along the whole range of return periods. These sites are Orestiada, Didymotycho, Sappes, and Ferres, which are characterized by low ground motion intensity. A common characteristic of these sites is that the higher ground motion values for them are provided by the area source model, SHARE_Area. The third group of sites, depicted by the black curves, exhibit moderate values of standard deviation. The island of Samothraki stands out among them due to its high ground motion values and the relatively large standard deviation of PGA and PGV for low return periods.

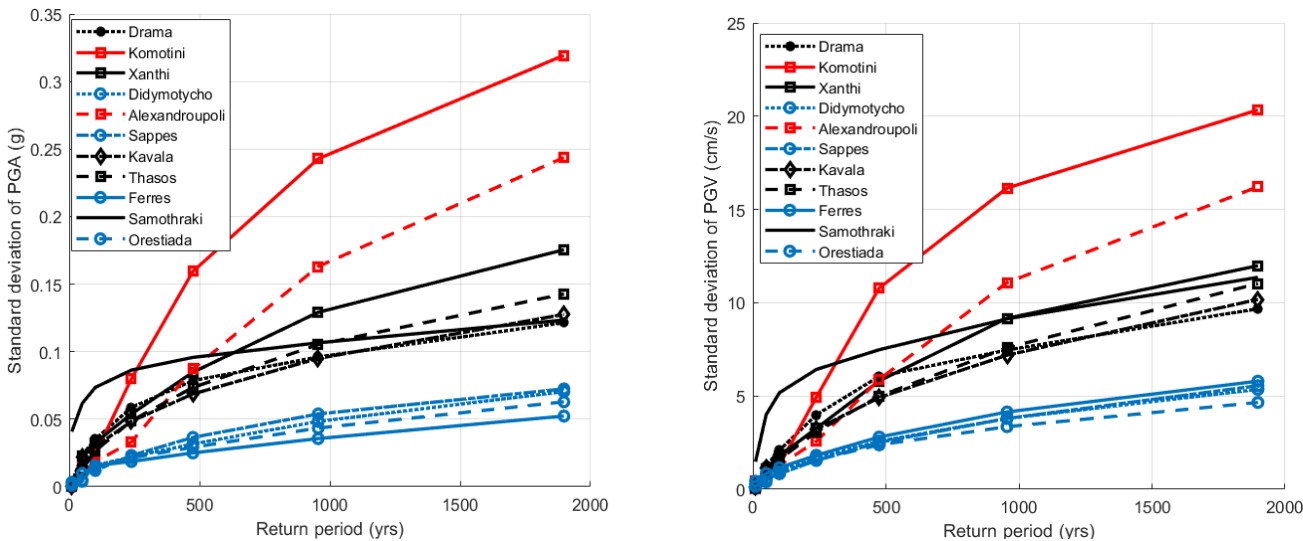

**Figure 7.** Standard deviation of PGA (**left**) and PGV (**right**) due to the seismic source model with respect to the return period for eleven sites in REMTH.

The aforementioned grouping of sites facilitates the interpretation of the variability of ground motion due to the source model across the whole return period range. If emphasis is put on the return period of 475 years, which corresponds to ground motion with a 10% probability of exceedance in 50 years and is associated with the seismic design and assessment of ordinary structures, then the city of Alexandroupoli could be moved to the group of moderate variability. For this group of sites, the standard deviation of PGA varies between 0.07 g and 0.096 g, whereas the standard deviation of PGV varies between 4.9 cm/s and 7.5 cm/s, with Kavala exhibiting the lower and Samothraki the higher values, respectively. Regarding the group of sites with the lowest variability, the standard deviation of PGA varies between 0.025 g and 0.036 g, and the one of PGV varies between 2.4 cm/s and 2.8 cm/s. On the other hand, the largest standard deviation for PGA is 0.16 g and for PGV is 10.8 cm/s, both in the city of Komotini.

Figure 8 presents the uniform hazard 5% damped acceleration response spectra (UHS) for the eleven cities and towns of REMTH for a return period equal to 475 years. For most of the sites shown in Figure 8 (Drama, Xanthi, Didymotycho, Sappes, Kavala, Thasos, and Orestiada), the area source model, SHARE_Area, provides the highest spectral accelerations. Similar spectral accelerations to SHARE_Area are predicted for Drama and Xanthi when PZ01_SHAREB is implemented. For the latter seismic source model and the city of Komotini, significantly higher spectral accelerations are predicted compared with the rest of the source models. For the city of Alexandroupoli, the TAP faults_SHAREB source model provides much higher spectral accelerations compared with the rest of the source models. The same source model, along with SHARE_FB, provides relatively low spectral accelerations for Komotini, Xanthi, and Drama. Overall, the variability of ground motion in terms of spectral accelerations due to the source model is significant.

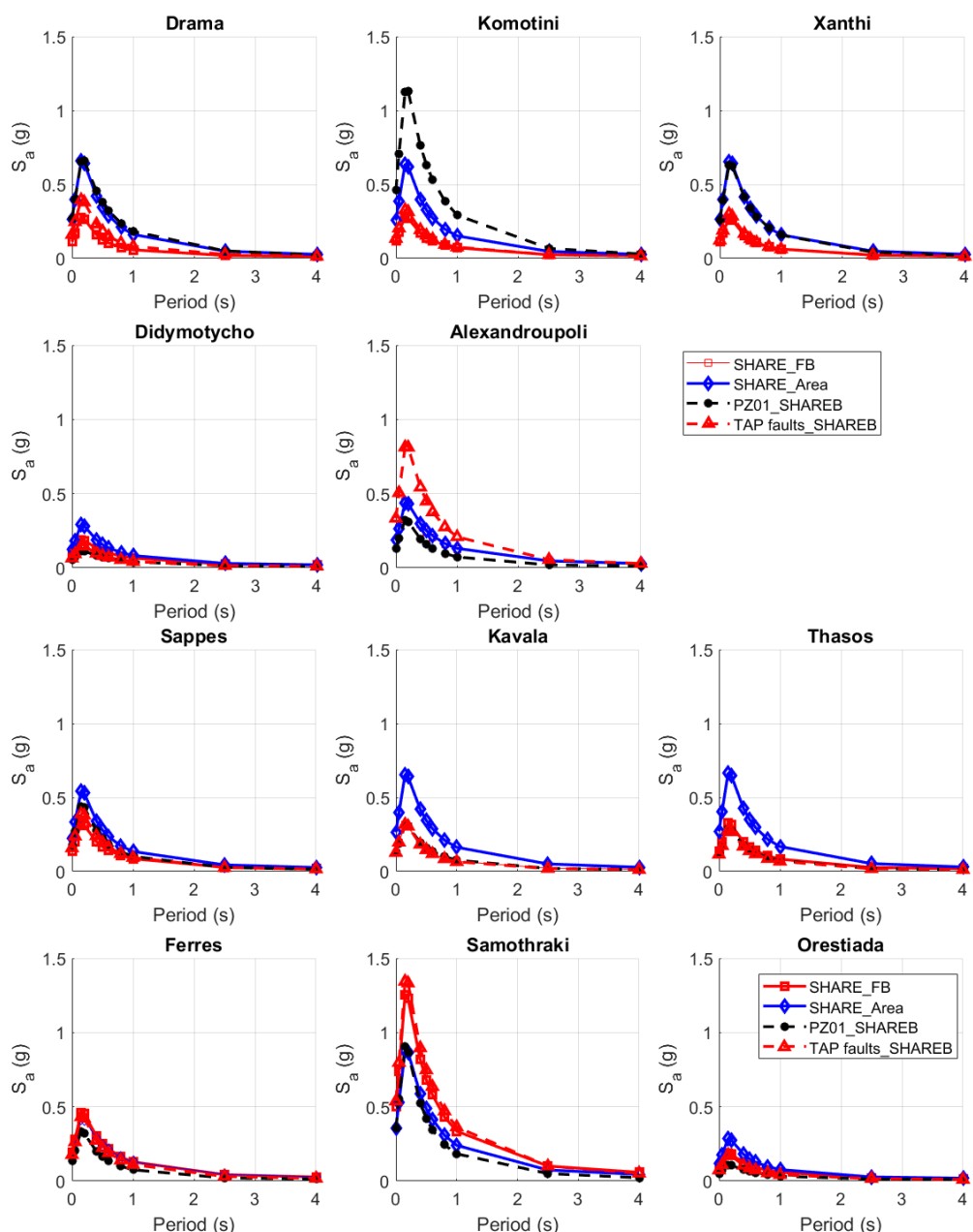

**Figure 8.** PSHA results for eleven important sites of REMTH, namely Drama, Komotini, Xanthi, Didymotycho, Alexandroupoli, Sappes, Kavala, Thasos, Ferres, Samothraki and Orestiada, in terms of uniform hazard acceleration response spectra for return period of 475 years.

*3.2. Final PSHA for REMTH and Comparison to Other Models*

The variability of ground motion due to the seismic source model is significant, as shown in Section 3.1. Within the framework of the Risk and Resilience Assessment Center—RISKAC REMTH project (MIS 5047293), the seismic hazard assessment of the whole region of REMTH is assessed. The authors' intention was to include the uncertainty associated with both the seismic source model and GMPE selection. The latter is considered to be adequately covered since the utilized GMPEs have been proven appropriate for implementation in Greece. It was deemed appropriate that the final PSHA logic tree should include both distributed seismicity and fault-based seismic source models. Moreover, the authors thought to consider seismicity models computed through observed seismicity and geological slip rates. Therefore, the final selection of seismic source models included the SHARE_Area, PZ01_SHAREB, and TAP faults_SHAREB models. The SHARE_Area was

the only distributed seismicity model considered, and its seismicity model came from the observed seismicity. The PZ01_SHAREB model includes seismic faults whose seismicity has been calculated from observed seismicity as well. The TAP faults_SHAREB model is a fault-based model with seismicity characteristics that were estimated through geological slip rates.

The neglection of SHARE_FB from the final PSHA logic tree was based on the following reasons:

- It exhibits the largest residual between its earthquake forecast and the observed seismicity of the whole region, as shown in Section 2 and Figure 4.
- Its features are similar to the TAP faults_SHAREB model in that they are both fault-based models and their seismicity is estimated through geological slip rates.
- The PSHA results shown in Figures 5, 6 and 8 indicate that the SHARE_FB model provides similar or lower ground motion intensity values compared with the TAP faults_SHAREB model. Thus, the latter source model was preferred to provide more conservative results.

However, the value of the SHARE_FB model is highly recognized by the authors and is taken into account in the subsequent seismic hazard disaggregation analyses. The final logic tree implemented in the PSHA for REMTH is shown in Figure 9.

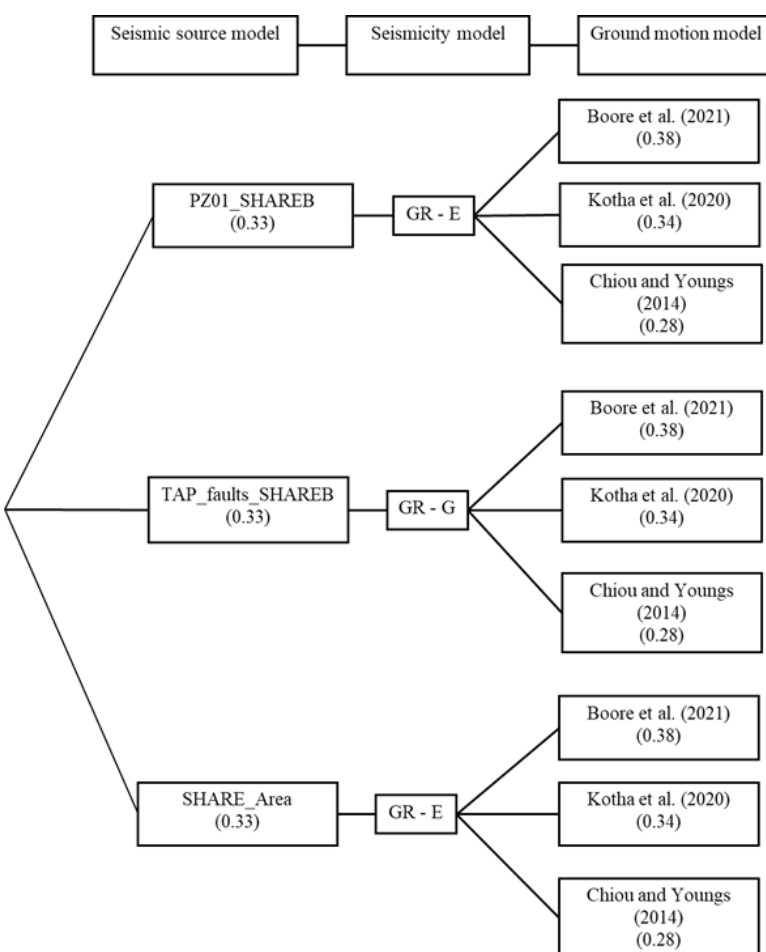

**Figure 9.** Final logic tree implemented for the PSHA of REMTH. GR-E and GR-G in the seismicity model denote that the Gutenberg–Richter distribution of the sources has been computed based on observed earthquake seismicity and geological slip rates, respectively [32,37–39].

It should also be noted that the source models that have been included in the recent ESHM20 model [23] were not considered, as the Risk and Resilience Assessment Center—RISKAC REMTH project had already advanced significantly by the time ESHM20 was published.

Figures 10 and 11 present the seismic hazard maps for REMTH for 475 and 955 years, respectively. The figures include the results of the present study (Figures 10a and 11a), which are compared with the results of the ESHM13 (Figures 10b and 11b) and ESHM20 (Figures 10c and 11c). ESHM13 and ESHM20 are pan-European seismic hazard models that have been developed based on seismicity and geological data, as well as GMPEs that have been tested on a European scale. Some of the seismic source models of ESHM13, namely the SHARE_Area and the background seismicity area sources of PZ01_SHAREB and TAP faults_SHAREB, were adopted in the present study. However, the difference between the present study and ESHM13 and ESHM20 lies in the fact that the components adopted herein were verified for the study area, whereas local data for seismic faults were considered as well. Moreover, the GMPEs that are used in the present study have been verified against strong local motion data.

Figures 10 and 11 show that except for the islands of Samothraki and Alexandroupoli, ESHM13 and ESHM20 present an almost uniform seismic hazard across REMTH. Furthermore, the PGA values of ESHM20 seem significantly lower than those of ESHM13. On the contrary, the results of the present study suggest local outbreaks of seismic hazards around Xanthi, Drama, and Komotini, highlighting the effect of local seismic faults. In addition, a zone of low seismic hazard is indicated at the north-eastern part of REMTH, which may not be found in ESHM13 and ESHM20. In Figure 12, the spatial distribution of the ratio between the PGA map, proposed in the present study, and ESHM13 is presented for a return period of 475 (Figure 12a) and 955 years (Figure 12b). The percentage shown in parenthesis within the legend of each figure denotes the percentage of the area belonging to each bin over the total area of the REMTH. Hence, for a return period of 475 years (Figure 12a), the results of the present study predict significantly higher PGA values (>20%) than ESHM13 over 18% of the region of REMTH, whereas moderate overprediction (10–20%) is apparent for 9% of the area of REMTH. Significantly lower PGA values (>20%) than ESHM13 are predicted by the present study for 19% of REMTH, while a similar percentage stands for moderate (10–20%) underprediction as well. For the rest of the region of REMTH (35%), the PGA values are similar between the present study and ESHM13, with differences being less than 10%. The situation is significantly altered for a return period of 955 years (Figure 12b). For most of REMTH, the PGA values of the present study are significantly lower (>20%) than the results of ESHM13. A similar analysis is presented in Figure 13, where the spatial distribution of the ratio between the PGA map, proposed in the present study, and ESHM20 is shown for a return period of 475 (Figure 13a) and 955 years (Figure 13b). It is noteworthy that for most of the study area (43% for 475 years and 39% for 955 years), the present study predicts notably higher PGA values than ESHM20. Moreover, the PGA ratios shown in Figure 13a,b do not change significantly with the increase in the return period, contrary to the comparison shown in Figure 12.

The design ground accelerations for rock site conditions that are prescribed by the Greek code [13] are uniformly distributed in REMTH, except for Samothraki, for which an increased value is suggested. More specifically, for the design of ordinary structures, which are designed based on a 475-year return period hazard level, the design ground acceleration is equal to 0.24 g for Samothraki and 0.16 g for the rest of REMTH. These design values are significantly lower than the results of the present study, with differences reaching up to 100%.

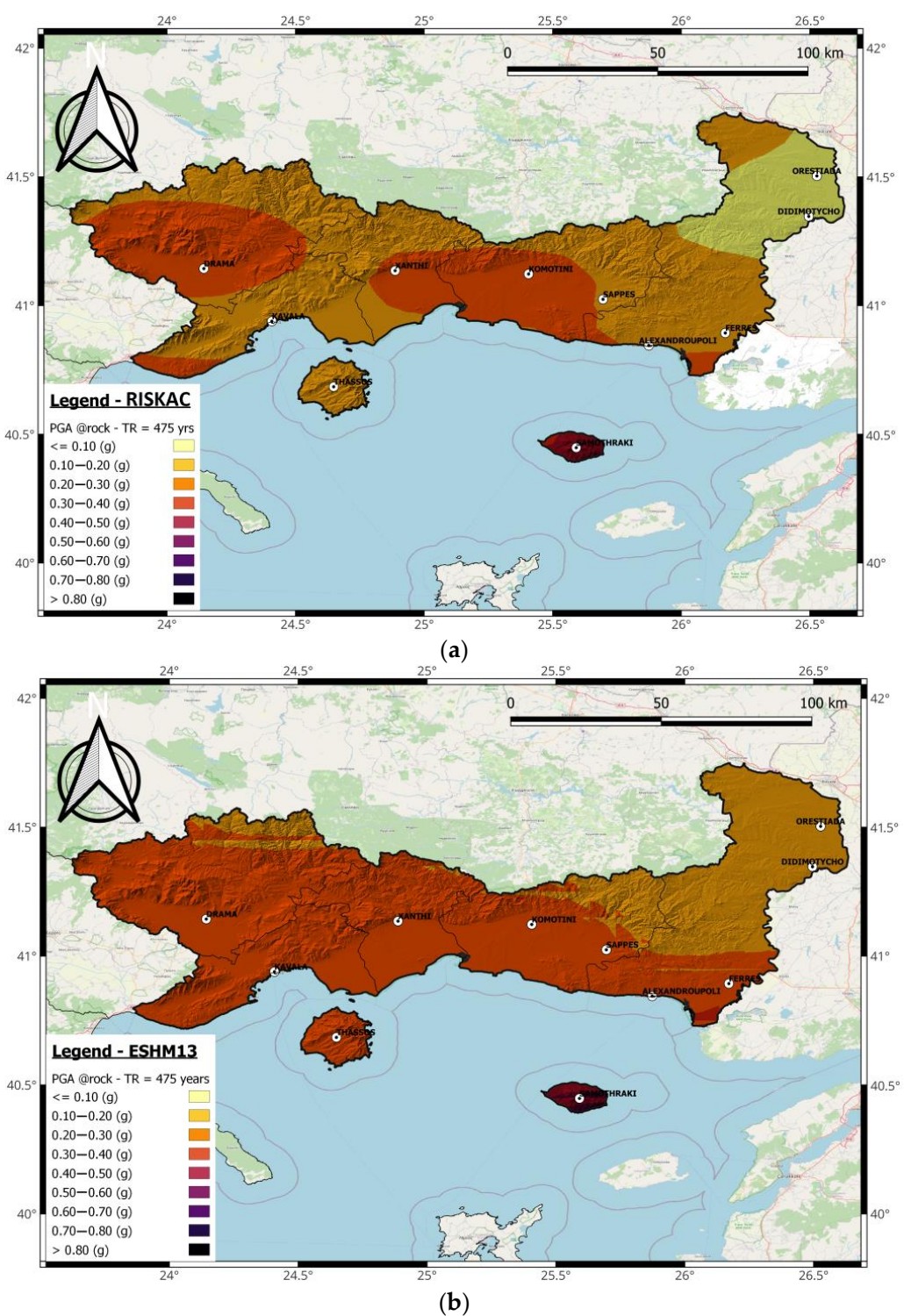

**Figure 10.** *Cont.*

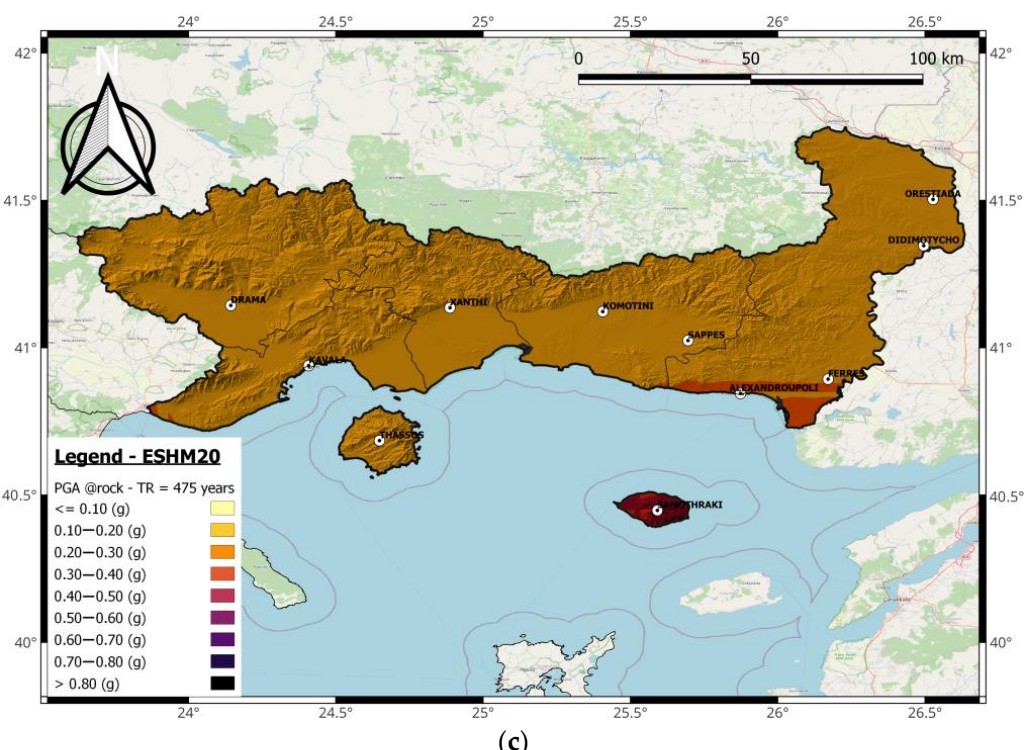

(**c**)

**Figure 10.** Seismic hazard map for PGA and a return period of 475 years according to (**a**) the present study, (**b**) ESHM13, and (**c**) ESHM20.

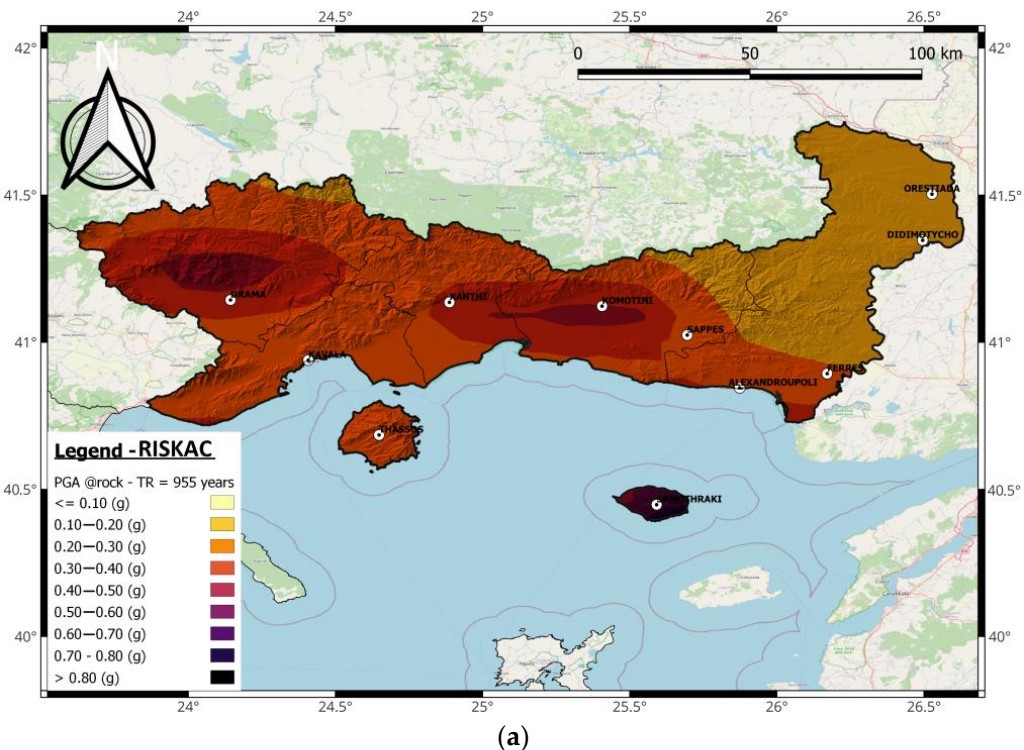

(**a**)

**Figure 11.** *Cont.*

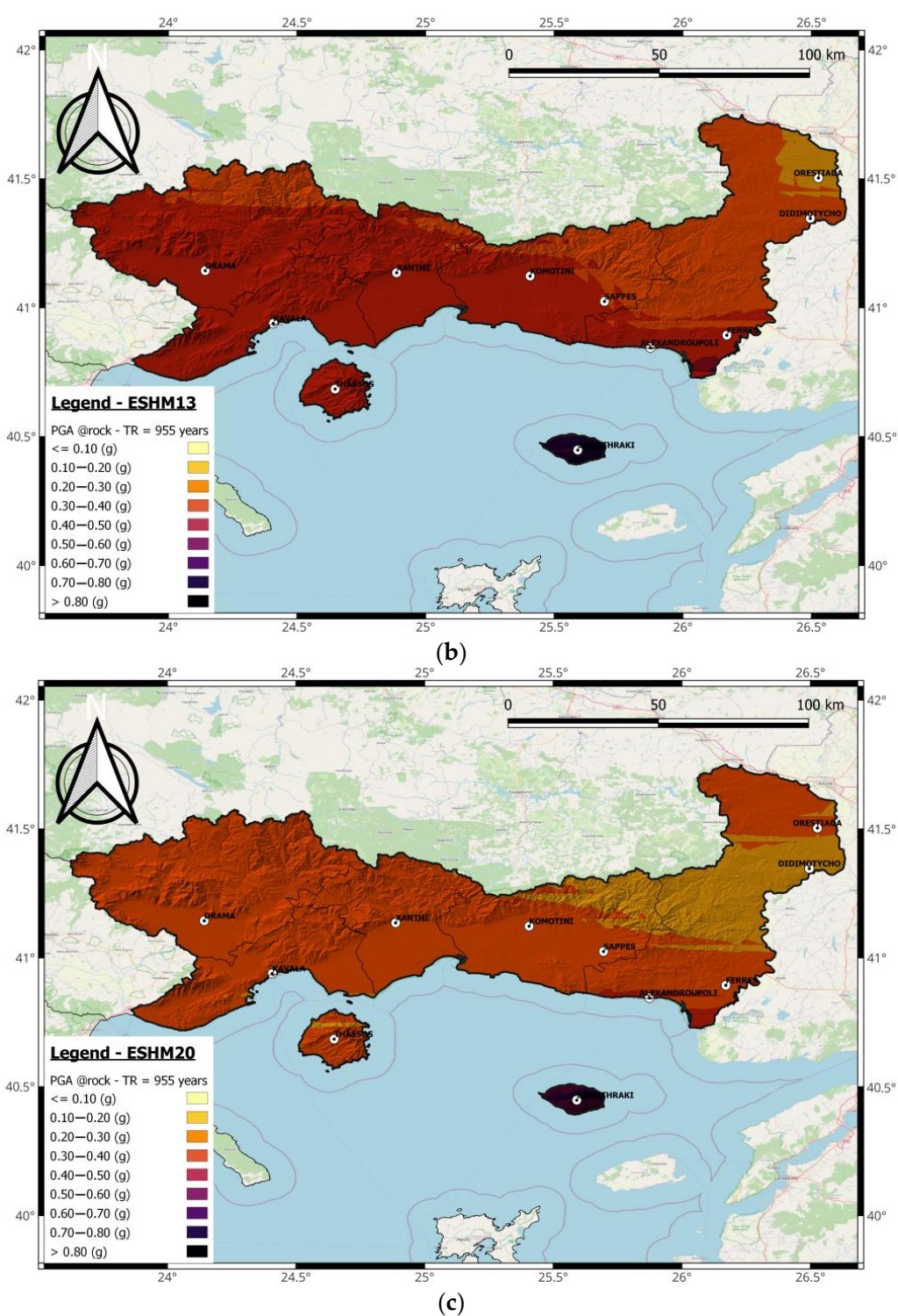

**Figure 11.** Seismic hazard map for PGA and a return period of 955 years according to (**a**) the present study, (**b**) ESHM13, and (**c**) ESHM20.

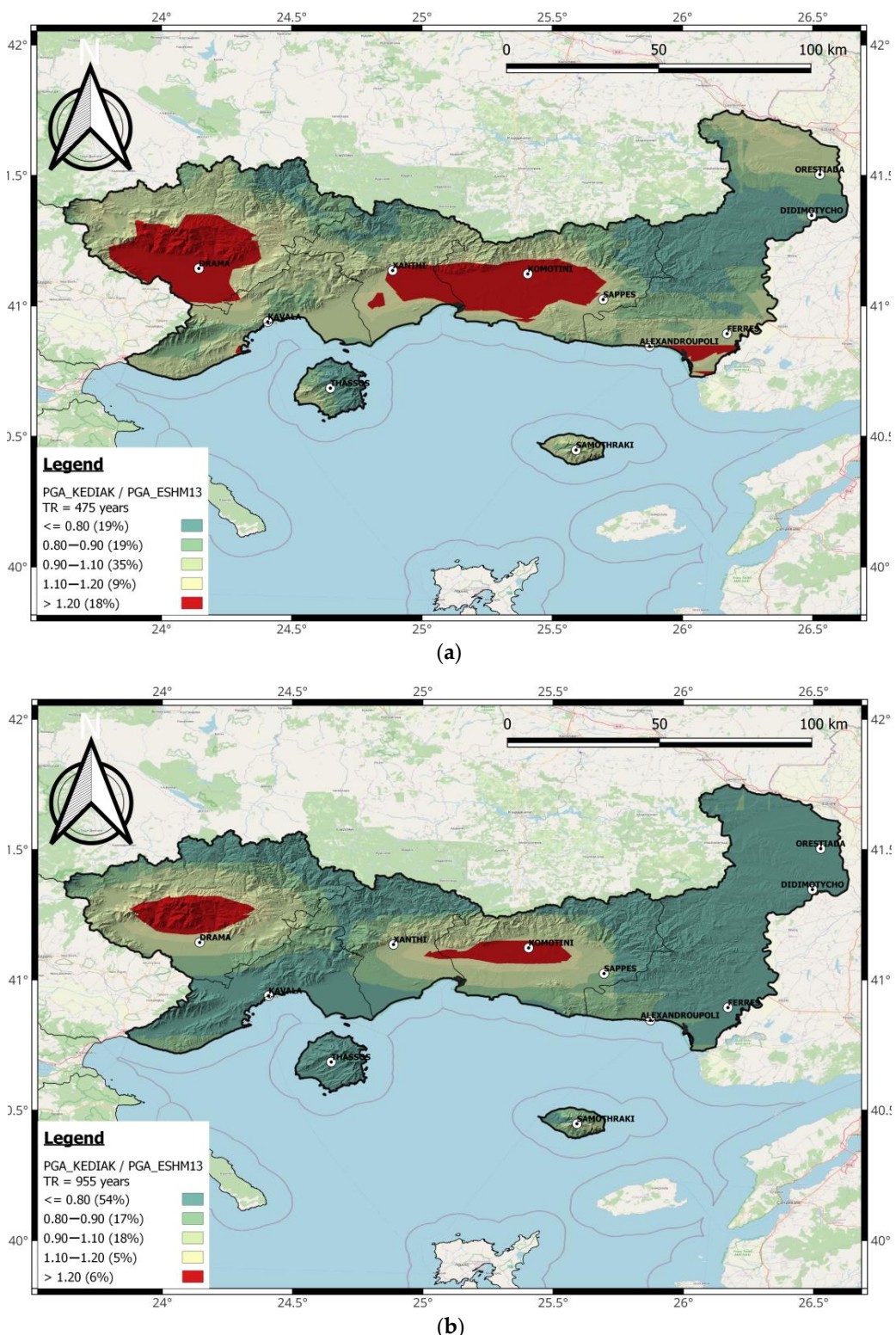

**Figure 12.** Spatial distribution of the ratio between the PGA of the present study and ESHM13 for a return period of (**a**) 475 and (**b**) 955 years.

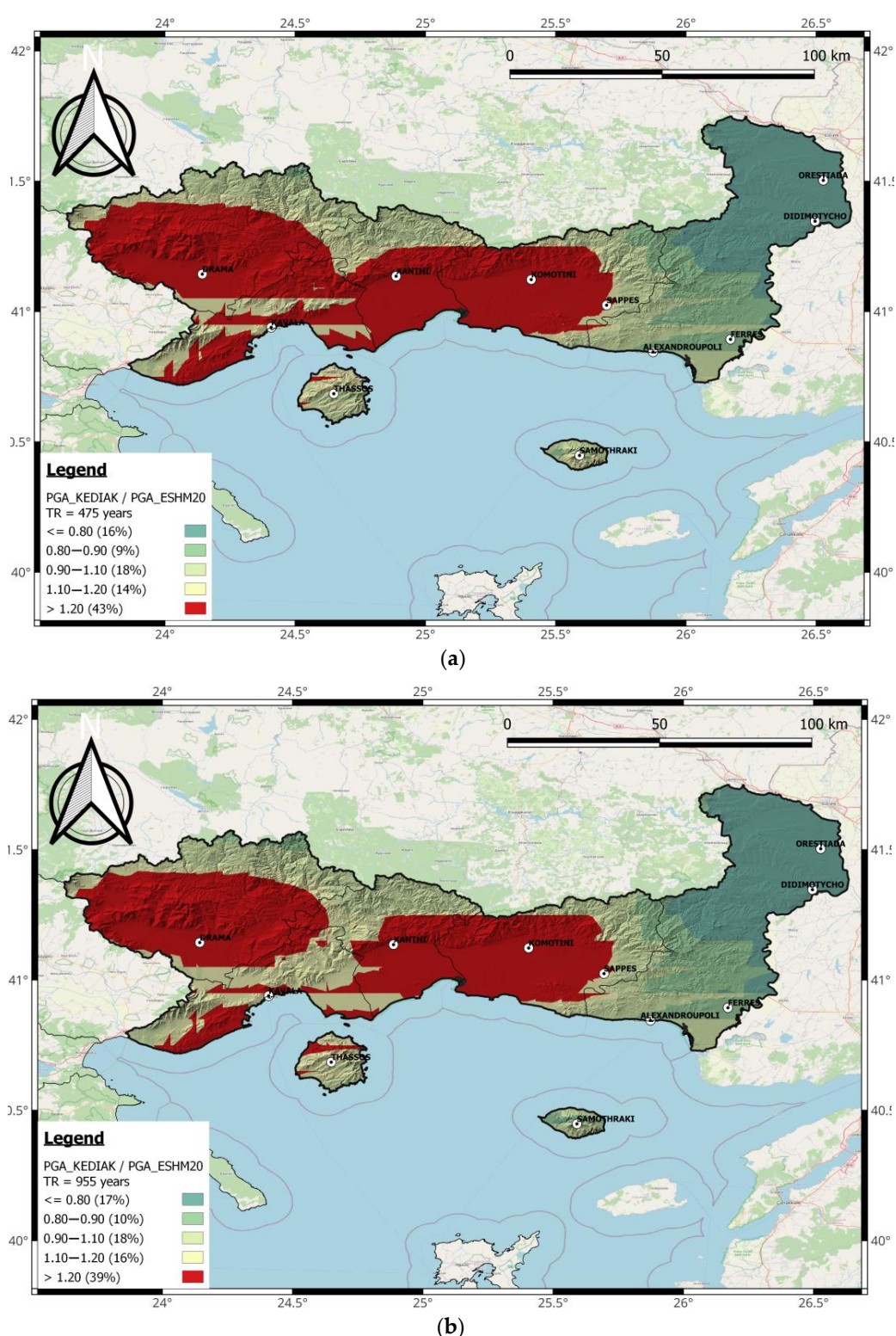

**Figure 13.** Spatial distribution of the ratio between the PGA of the present study and ESHM20 for a return period of (**a**) 475 and (**b**) 955 years.

Figures 14 and 15 present the comparison between the UHS of six important cities (in terms of population, seismicity, and tourism) of REMTH predicted by the present study, ESHM13 and ESHM20, and the design acceleration response spectrum of Eurocode 8 [41], for a return period of 475 and 955 years, respectively. For Drama, Komotini, and Xanthi, the present study predicts amplified acceleration response spectral with respect to ESHM13

and ESHM20, whereas for Samothraki and Alexandroupoli, its predictions fall beneath them. Furthermore, the present study's UHS for Kavala lies between that of ESHM13 and that of ESHM20. Moreover, it is noted that the ESHM20 estimates are always less conservative than the present study and ESHM13, with the exception of Samothraki and Alexandroupoli. Also, with the exception of Samothraki, for every other site shown in Figures 14 and 15, the UHS of ESHM13 is always larger than ESHM20. Nevertheless, for most of the sites, the differences in spectral accelerations are not that significant. The design acceleration response spectrum of Eurocode 8, which is applied for new structures in Greece, falls significantly below the present study's UHS for Samothraki. However, for the rest of the cities shown in Figures 14 and 15, the resemblance of the design code's spectrum to the results reported herein seems adequate, especially for long periods. A major underestimation of spectral accelerations is noted for low periods, that is, from PGA and up to the end of the plateau region of the code's spectrum. The differences seem to be amplified for larger return periods, as depicted in Figure 15.

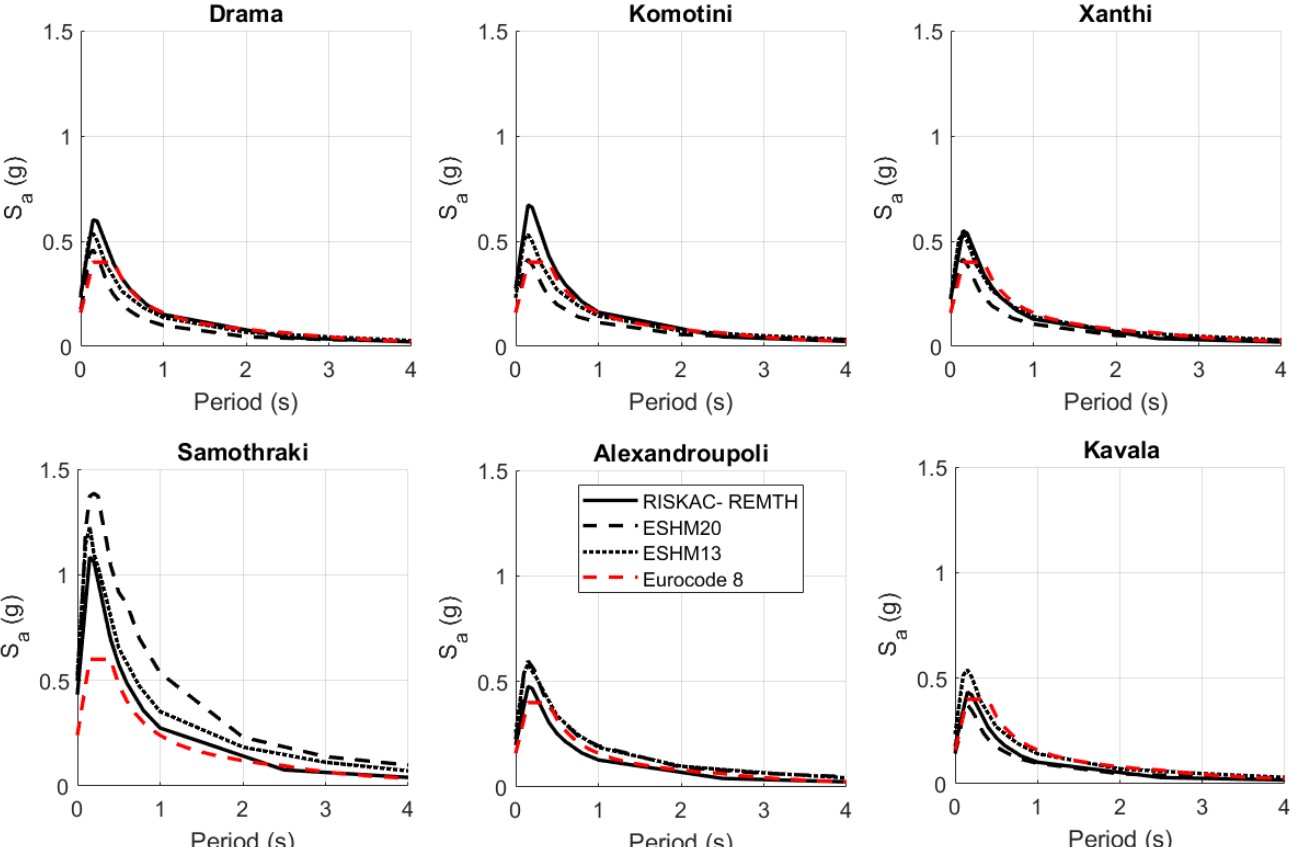

**Figure 14.** Uniform hazard acceleration response spectra (5% damped) of six important cities in REMTH, namely Drama, Komotini, Xanthi, Samothraki, Alexandroupoli and Kavala, according to the present study (RISKAC—REMTH) and the European seismic hazard models ESHM13 and ESHM20 for a return period of 475 years.

The differences between the PSHA results of RISKAC, ESHM13, and ESHM20, which were described above, are attributed to the differences between the various components of these hazard models. The RISKAC and the ESHM13 have been developed following a similar rationale. The RISKAC model uses the same area sources as ESHM13, as well as the same background seismicity sources in combination with seismic faults. An important difference between them in terms of seismic sources arises from the fact that within RISKAC, local seismic faults are included that are responsible for the higher PGA and PGV values around Drama, Xanthi, and Komotini. Moreover, the GMPEs used in ESHM13 were

validated based on data-driven procedures and expert judgment through comparison with strong motion data from all of Europe. On the other hand, the GMPEs chosen for the RISKAC model have been validated against strong motion data from Greece only. The rationale for the development of ESHM20 differs from the other two. In terms of source modeling, the basic difference comes from the fact that no background seismicity area sources are used in combination with the seismic faults. Instead, a smoothed gridded seismicity model is used. Moreover, in ESHM20, the magnitude–frequency distribution of seismic sources is described through the Gutenberg–Richter (GR) distribution, which is also used in ESHM13 and RISKAC. Additionally, ESHM20 incorporates the pareto distribution, which causes a reduction of higher-magnitude events compared with the GR distribution. The ground motion in ESHM20 is modeled through the concept of a scaled backbone ground motion model logic tree. That is, a single ground motion model is calibrated, and to this model, adjustment factors are applied that quantify the uncertainty in the expected ground motion as a result of the limited knowledge of the seismological properties in a region. The fault type is not considered in the GMPE of Kotha et al. (2020;2022), which is adopted in ESHM20, but region-specific adjustments are made in terms of source-region-specific scaling and anelastic distance attenuation.

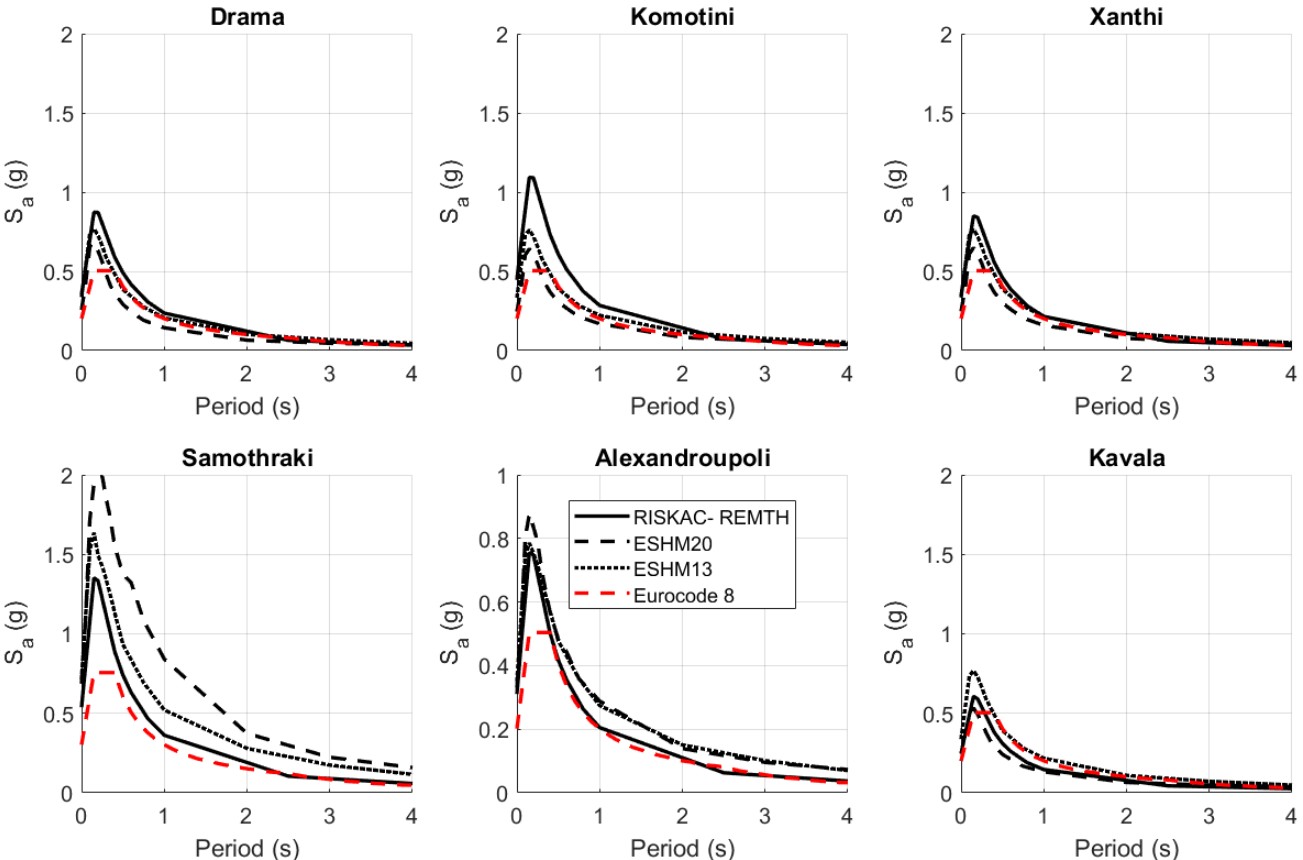

**Figure 15.** Uniform hazard acceleration response spectra (5% damped) of six important cities in REMTH, namely Drama, Komotini, Xanthi, Samothraki, Alexandroupoli and Kavala, according to the present study (RISKAC—REMTH) and the European seismic hazard models ESHM13 and ESHM20 for a return period of 955 years.

### 3.3. Disaggregation of the Seismic Hazard and Earthquake Scenarios

Following the PSHA, the disaggregation of seismic hazards in terms of magnitude (M), distance (R), and ground motion variability, epsilon ($\varepsilon$), is undertaken. In PSHA, all the seismic sources that contribute to the hazard at a site are considered and integrated. However, this representation of seismic hazards may not be convenient in cases where

specific seismic scenarios are required, as, for example, in studies where time series of ground motions are needed. The implementation of seismic hazard disaggregation leads to the identification of the seismic scenario, in terms of M-R-$\varepsilon$, that contributes the most to the overall seismic hazard at a site. In the present study, the recommendations and approaches of [42] were followed to disaggregate the seismic hazard for the six cities of REMTH shown in Figures 14 and 15, for a return period equal to 475 and 955 years. The disaggregation was performed for the intensity measure levels, which were calculated through PSHA, separately for each seismic source model shown in Table 1 to investigate the variability of the disaggregation results due to the source model. Furthermore, the disaggregation was performed for three intensity measures of ground motion, namely PGA, PGV, and Sa (0.15 s). The period of 0.15 s was deemed representative of the fundamental period of the buildings found in the examined cities, according to the census provided by the Hellenic Statistical Authority [43]. Tables S1–S12, in the electronic supplement, present the seismic scenarios that mostly contribute to the seismic hazard of the examined sites with respect to the seismic source model (SM) and the intensity measure (IM), for a return period of 475 and 955 years, respectively. It should be noted that R refers to the Joyner–Boore distance ($R_{JB}$).

Figure 16 presents the seismic scenarios with the largest contribution to hazard based on PGA, in terms of M, R, and epsilon, for six major cities of REMTH, for two return periods (TR), with respect to the adopted seismic source model. Regarding the earthquake magnitude (Figure 16a,b), some significant differences between the various source models are observed, especially for the return period of 475 years. For example, for Drama, the earthquake magnitude of the scenarios with the highest contribution to hazard varies between 5.1 and 7.1, depending on the adopted source model. Large differences are observed for Komotini and Xanthi as well. The lowest earthquake magnitude for these sites comes from the SHARE_FB source model, whereas the rest of the models provide similar results. On the other hand, for Alexandroupoli, the various source models agree with each other adequately, whereas for Kavala, as for all source models, earthquake magnitude is denoted as the one with the highest contribution. The differences in earthquake magnitude among the source models seem to be smoothed for the return period of 955 years, with a few exceptions. Regarding the distance between the site of interest and the earthquake rupture (Figure 16c,d), differences are also apparent for both return periods. For most sites, the PZ01_SHAREB and the TAP faults_SHAREB source models provide the lowest distance, whereas the SHARE_FB results in the largest one. A similar observation stands for epsilon as well (Figure 16e,f). Adoption of the source model TAP faults_SHAREB results in the lowest epsilon value, which is usually very close to zero, meaning that the estimated ground motion is closer to the mean estimate of the selected GMPEs.

Regarding the variability of characteristics of the seismic scenarios with the highest contribution to hazard with respect to the selected intensity measure, significant variations were found only for specific sites and source models, such as Kavala and SHARE_FB. On the other hand, for the TAP faults_SHAREB source model and most of the considered sites, practically no differences in the most contributing seismic scenarios were observed among the intensity measures. Nevertheless, the largest differences were observed in epsilon rather than M or R. The final selection of the representative seismic scenarios for each site, among the scenarios shown in Tables S1–S12, was based on the consistency of occurrence of a scenario, that is, if a scenario keeps coming up even if the source model or the intensity measures change. The final selection of the seismic scenarios for each city in REMTH is given in Table 3.

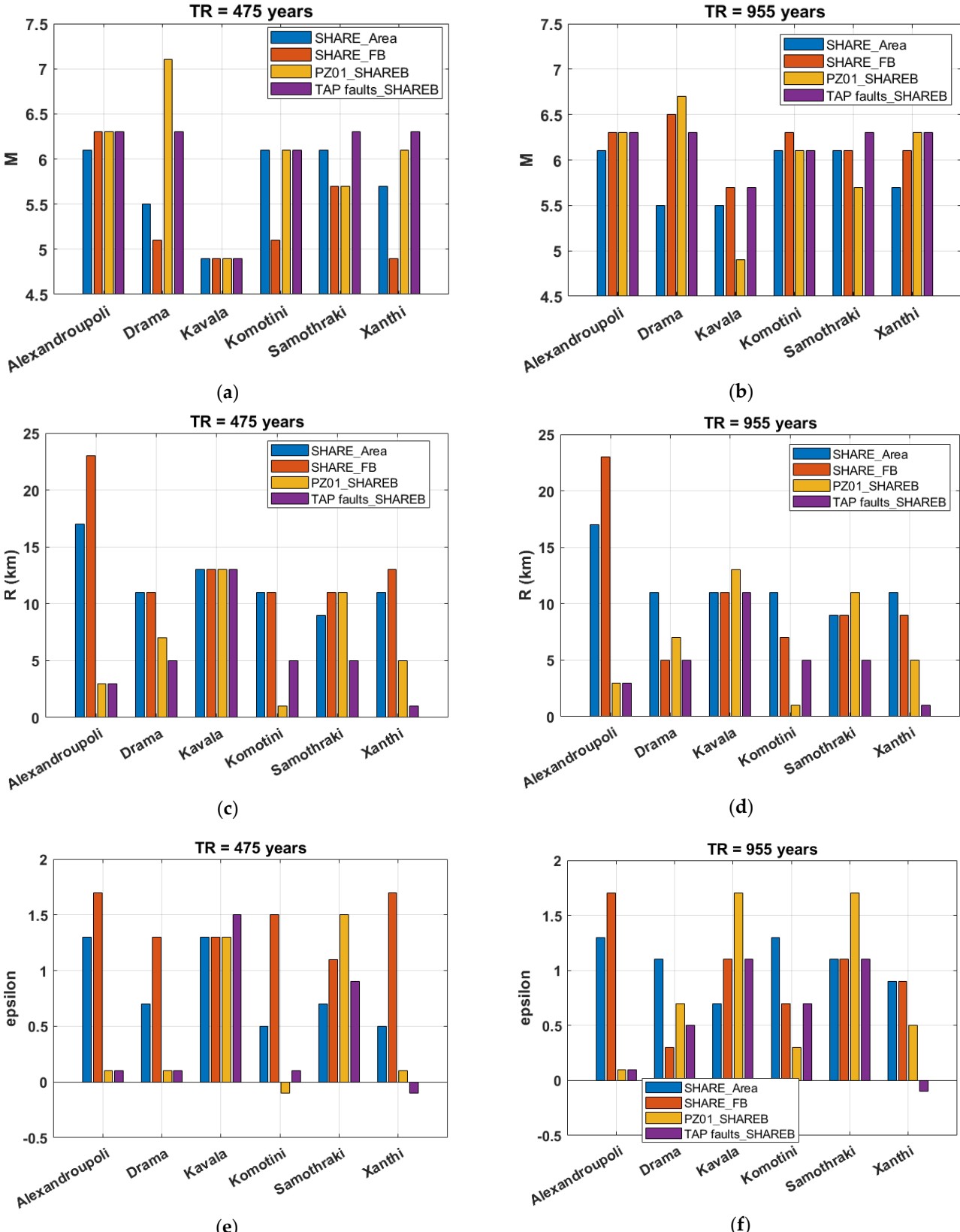

**Figure 16.** Seismic scenarios with the largest contribution to hazards, based on PGA, in terms of M, R, and epsilon for the six great cities of REMTH with respect to the seismic source model. (**a,c,e**) refer to a return period of 475 years, while (**b,d,f**) refer to a return period of 955 years.

**Table 3.** Final selection of representative seismic scenarios based on the disaggregation results.

| City | TR (Years) | M | R (km) | ε |
|------|-----------|---|--------|---|
| Alexandroupoli | 475 | 6.3 | 3.0 | 0.1 |
| | 955 | 6.3 | 3.0 | 0.1 |
| Drama | 475 | 6.3 | 5.0 | 0.1 |
| | 955 | 7.1 | 7.0 | 0.3 |
| Kavala | 475 | 5.7 | 11.0 | 0.7 |
| | 955 | 5.7 | 11.0 | 1.1 |
| Komotini | 475 | 6.1 | 1.0 | −0.1 |
| | 955 | 6.1 | 1.0 | 0.7 |
| Samothraki | 475 | 6.3 | 5.0 | 0.9 |
| | 955 | 6.3 | 5.0 | 1.1 |
| Xanthi | 475 | 6.3 | 1.0 | −0.1 |
| | 955 | 6.3 | 1.0 | 0.1 |

*3.4. Deterministic Seismic Hazard Assessment (DSHA) and Selection of Ground Motion Recordings*

Based on the selected seismic scenarios, which are presented in Table 3, a DSHA was undertaken. DSHA included the calculation of the acceleration response spectrum (for a 5% damping ratio) for specific earthquake scenarios and the sites shown in Table 3. Towards this aim, the GMPEs shown in Table 2 were implemented. To supplement the deterministic response spectra, real ground motion recordings were gathered from Greek [33] and international ground motion databases [44,45], with seismotectonic features similar to the selected earthquake scenarios. No scaling of the ground motions was implemented, so the intensity and frequency content of the motions were not modified. The criteria used for the search of ground motions were the magnitude (M), the site-to-source distance ($R_{jb}$), the local site conditions (rock) represented by the average shear wave velocity of the upper 30 m of soil ($V_{S30}$), as well as the rupture mechanism (wherever this was possible). In addition, another important criterion was the resemblance between the ground motion response spectra and the response spectra computed through the DSHA. Table 4 includes the ground motions selected for the six cities of REMTH, while Figure 17 presents their acceleration response spectra along with the response spectra from DSHA. In Table 4, PGArotD50 is the peak ground acceleration of the rotD50 component of the ground motion [32].

**Table 4.** Selected ground motions, consistent with the representative seismic scenarios and the DSHA response spectra for the six cities of REMTH.

| # | RSN | DDMMYY HHMMSS | Latitude (°) | Longitude (°) | M | $R_{jb}$ (km) | $V_{S30}$ (m/s) | Mechanism | Station Code | PGArotD50 (g) | Source |
|---|-----|---------------|--------------|---------------|---|---------------|-----------------|-----------|--------------|---------------|--------|
| 1 | 861 | 07/09/1999 11:56:51 | 38.06 | 23.54 | 6.0 | 4.2 | 582 | Normal | ATH3 | 0.290 | 1 * |
| 2 | 860 | 07/09/1999 11:56:51 | 38.06 | 23.54 | 6.0 | 2.4 | 412 | Normal | SPLB | 0.337 | 1 |
| 3 | 599 | 24/02/1981 20:53:36 | 38.07 | 23.00 | 6.7 | 5.3 | 339 | Normal | KORA | 0.257 | 1 |
| 4 | 786 | 17/05/1995 04:14:25 | 40.1526 | 21.6183 | 5.4 | 10.4 | 470 | Normal | 1CHR | 0.128 | 1 |
| 6 | 230 | 25/05/1980 | 37.677 | 118.900 | 6.06 | 1.1 | 382 | Normal | Convict Creek | 0.440 | 2 ** |
| 7 | - | 22/10/1999 02:18:58 | 23.445 | 120.506 | 5.9 | 8.3 | 877 | - | TW | 0.500 | 3 *** |
| 8 | - | 24/08/2016 01:36:32 | 42.6983 | 13.2335 | 6.0 | 2.0 | 498 | Normal | NRC | 0.367 | 3 |
| 9 | - | 18/01/2017 10:25:24 | 42.53115 | 13.29272 | 5.4 | 10.4 | 670 | Normal | AMT | 0.137 | 3 |
| 10 | | 07/04/2009 17:47:37 | 42.303 | 13.486 | 6.6 | 11.1 | 696 | Normal | AQG | 0.120 | 3 |
| 11 | 4369 | 10/6/1997 | 43.045 | 12.835 | 5.5 | 11.7 | 694 | Normal | "Nocera Umbra-Salmata" | 0.187 | 2 |
| 12 | 4509 | 07/04/2009 | 42.275 | 13.464 | 5.6 | 11.1 | 685 | Normal | "L'Aquila—V. Aterno -Colle Grilli" | 0.150 | 2 |
| 13 | - | 24/08/2016 01:36:32 | 42.6983 | 13.2335 | 6.0 | 1.4 | 670 | Normal | AMT | 0.387 | 3 |
| 14 | - | 30/10/2016 06:40:18 | 42.83794 | 13.12324 | 6.6 | 12.8 | 698 | Normal | CSC | 0.163 | 3 |
| 15 | 4099 | 28/92004 | 35.818 | 120.366 | 6.0 | 3.2 | 523 | Strike-slip | "Parkfield—Cholame 2E" | 0.485 | 2 |

**Table 4.** *Cont.*

| # | RSN | DDMMYY HHMMSS | Latitude (°) | Longitude (°) | M | $R_{jb}$ (km) | $V_{S30}$ (m/s) | Mechanism | Station Code | PGArotD50 (g) | Source |
|---|-----|---------------|--------------|---------------|---|---------------|-----------------|-----------|--------------|---------------|--------|
| 16 | 4122 | 28/9/2004 | 35.818 | 120.366 | 6.0 | 4.7 | 511 | Strike-slip | "Parkfield—Gold Hill 3W" | 0.608 | 2 |
| 17 | 459 | 24/04/1984 | 37.310 | 121.679 | 6.2 | 9.9 | 663 | Strike-slip | "Gilroy Array #6" | 0.268 | 2 |
| 18 | 4064 | 28/9/2004 | 35.818 | 120.366 | 6.0 | 4.3 | 657 | Strike-slip | "PARKFIELD—DONNA LEE" | 0.341 | 2 |
| 19 | - | 26/10/2016 19:18:06 | 42.91014 | 13.1406 | 5.9 | 5.9 | 498 | Normal | NRC | 0.282 | 3 |

* 1 Margaris et al., 2021 [33], ** 2 PEER Ground Motion Database [44], *** 3 Engineering Strong Motion Database [45].

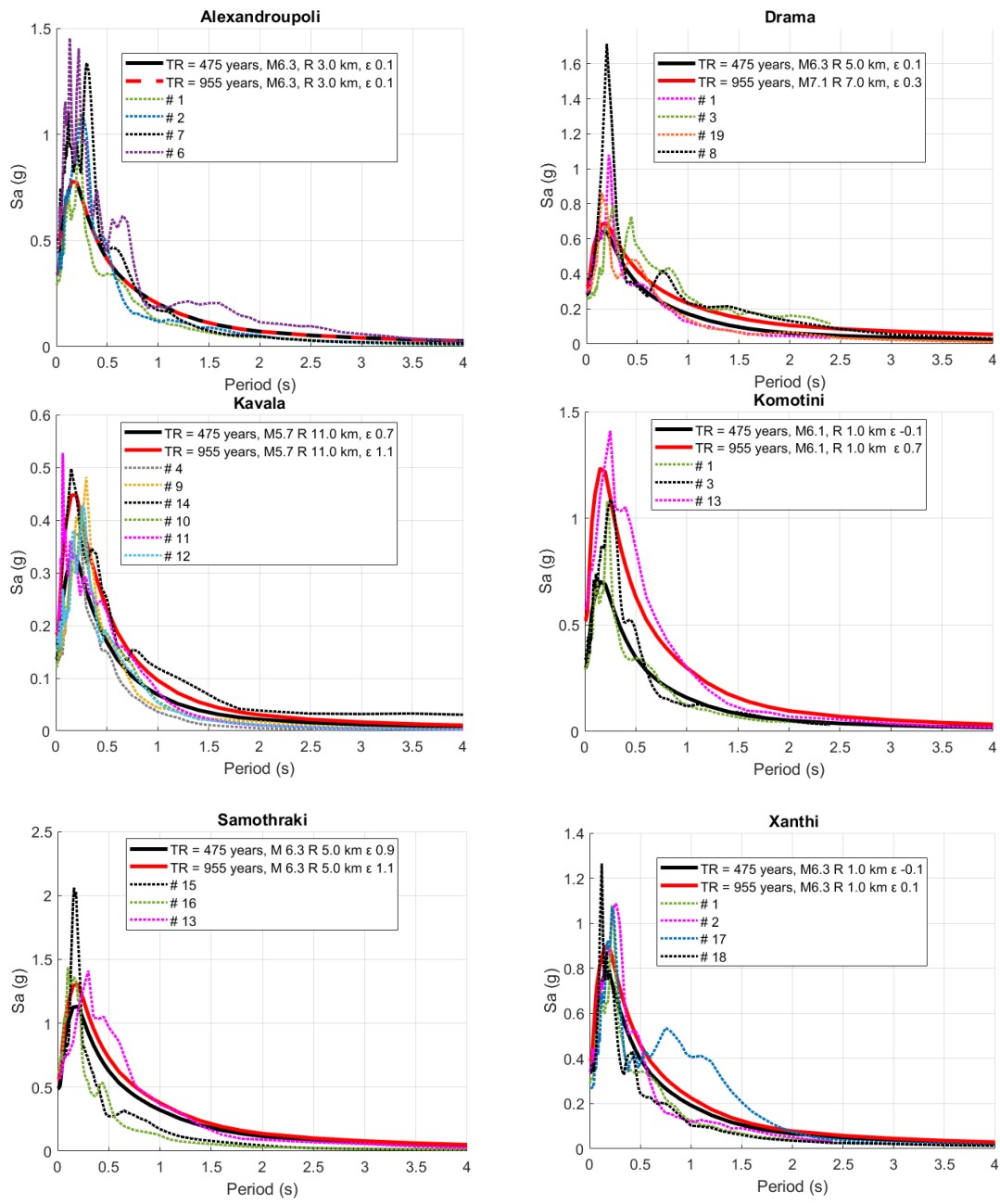

**Figure 17.** DSHA acceleration response spectra (5% damped) for the selected cities of REMTH, namely Alexandroupoli, Drama, Kavala, Komotini, Samothraki and Xathi, and response spectra of scenario-based compatible ground motions.

## 4. Discussion and Conclusions

In the present study, a comprehensive seismic hazard assessment is undertaken for the region of REMTH, taking into account modern seismic source models and GMPEs, which have been validated against local strong motion data. Towards this end, the method of PSHA is adopted and implemented for several return periods and ground motion intensity measures. The variability of PSHA results with respect to the selection of the source model has been examined. For most of the sites considered, the variability of PSHA results due to source model selection is significant in terms of seismic hazard curves. Noteworthy differences are observed in terms of the uniform hazard acceleration response spectra, or UHS.

A final logic tree is proposed for the PSHA of REMTH, based on criteria that include (a) the agreement between the earthquake forecast of the source models and the observed seismicity of the study area, (b) the intent to include seismicity models developed through different rationales, and (c) GMPEs that reproduce more accurately the local strong motion characteristics. PSHA maps were created and proposed based on the logic tree approach for several ground motion intensity measures and return periods, and a part of them is presented herein. The present study's results are compared against the results of the recent ESHM13 and ESHM20 seismic hazard models. With respect to the former model, the present study predicts similar PGA values for a return period of 475 years across a wide area of REMTH, whereas in the vicinity of large cities it predicts higher values, highlighting the effect of local seismic faults. On the other hand, for a return period of 955 years, the present study predicts significantly higher PGAs for a very limited area of REMTH, whereas for most of it, it predicts lower values than ESHM13. With respect to ESHM20, a relatively constant comparison can be observed between that and the present study, independent of the return period. For most of REMTH, the present study estimates much higher PGAs than ESHM20. Therefore, as a general comment, one could argue that the PSHA maps proposed herein, having the significant advantage of local verification of the PHSA components, stand between the ones proposed by ESHM13 and ESHM20. Moreover, significant differences are observed between the ESHM13 and ESHM20 maps.

Comparison between the UHS of the present study, the ESHM13 and ESHM20, for six important cities of REMTH did not reveal noteworthy differences for intermediate and long periods. Furthermore, the current seismic design code seems to adequately cover that period range. However, important differences are spotted for low periods, which are usually associated with the response of parts of geostructures and low-rise buildings, which compose most of the building stock in the region. If anything, the present study's results, along with other recent studies in Greece [9–12], highlight the need for revising the seismic hazard zones adopted in the seismic design code.

Supplementarily to PSHA, disaggregation of the seismic hazard for six large cities in REMTH was performed. The variability due to the seismic source model selection was also depicted in the estimation of the seismic scenario (in terms of M-R-ε), which contributes the most to the hazard of the considered sites. Multiple intensity measures were considered for the disaggregation of seismic hazards as well; however, the variability of those results was minor compared with the source model selection. Representative earthquake scenarios were chosen based on consistency criteria, and DSHA was conducted for the six cities of REMTH to complement the PSHA results. Additionally, scenario-based and acceleration spectrum-based compatible ground motions were gathered to provide a useful tool for local engineers and administrative authorities in the region to assess existing structures or design new ones.

**Supplementary Materials:** The following supporting information can be downloaded at: https://www.mdpi.com/article/10.3390/geohazards4030014/s1; Tables S1–S12: Most contributing seismic scenarios for the main cities of REMTH for a return period of 475 and 955 years.

**Author Contributions:** Conceptualization, D.S. and B.M.; methodology, D.S. and B.M.; investigation, D.S and B.M.; resources, I.M.D.; writing—original draft preparation, D.S.; writing—review and editing, B.M, N.K. and I.M.D.; supervision, B.M. and N.K.; project administration, I.M.D.; All authors have read and agreed to the published version of the manuscript.

**Funding:** We acknowledge the support of this work by the project "Risk and Resilience Assessment Center–Prefecture of East Macedonia and Thrace-Greece" (MIS 5047293), which is implemented under the Action "Reinforcement of the Research and Innovation Infrastructure", funded by the Operational Program "Competitiveness, Entrepreneurship and Innovation" (NSRF 2014–2020), and co-financed by Greece and the European Union (European Regional Development Fund).

**Institutional Review Board Statement:** Not applicable.

**Informed Consent Statement:** Not applicable.

**Data Availability Statement:** Not applicable.

**Conflicts of Interest:** The authors declare no conflict of interest.

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
