# Peer review of "Seismic Hazard in Greece: A Comparative Study for the Region of East Macedonia and Thrace"

_2624-795X, doi:10.3390/geohazards4030014_

Round 1

Reviewer 1 Report

The authors have effectively collected, analyzed, and well presented the data. The methods employed are applicable, appropriately applied, and adequately described. The manuscript is well-written and supplemented with informative maps and figures. It is in an advanced stage and does not require revision prior to publication (except a typo in "Table 2: Description of the Seismic source models considered in the present study" that is actually for GMM)

Author Response

We would like to thank the reviewer for his comments. The caption of Table 2 has been corrected accordingly.

Author Response

We would like to thank the reviewer for his/her comments and for the time he/she spent on it. Please, in the following find our response:

1) Originally, the seismic hazard curves calculated through Openquake, describe the probability of exceedance of specific PGA and PGV values P(y>Y) (see equation (1) of the manuscript). Utilizing equation (1) of the manuscript  and the fact that the inverse of the annual rate of exceedance, λy, is equalt to the return period, the return period is calculated as:

TR=-t/ln(1-poe). 

This explanation has been included in lines 264 - 270 of the revised manuscript.

2) An additional paragraph has been included in lines 462 - 486 of the revised manuscript to address the comment of the reviewer.

Reviewer 3 Report

In this manuscript (ms), the authors present a seismic hazard analysis for East Macedonia and Thrace (EMTH). The authors employ modern methods of probabilistic seismic hazard assessment (PSHA) for the region of EMTH and compare the results obtained by various seismic source models. They also present PSHA results from the most important cities of EMTH in a comparative way. Moreover, an updated seismic hazard map of the study area is proposed (see Figs. 10(a) and 11(a) for 475 and 955 years, respectively) and a comparative disaggregation analysis is performed to estimate the earthquake scenarios with the largest contribution to the seismic hazard (see Fig.17). The ms is extensive, well written, and the results, which are original, support the conclusions mentioned in the Section labeled “4. Discussion”. Hence, this research deserves publication in GeoHazards, but before publication the presentation needs to be improved in the following points:

1)In line 29, the authors mention various regional or local scale seismic hazard studies, however, they miss to mention the following recent related papers

Sarlis, N.V.; Skordas, E.S. Study in Natural Time of Geoelectric Field and Seismicity Changes Preceding the Mw6.8 Earthquake on 25 October 2018 in Greece. Entropy 2018, 20, 882. https://doi.org/10.3390/e20110882

Varotsos, P.K.; Perez-Oregon, J.; Skordas, E.S.; Sarlis, N.V. Estimating the Epicenter of an Impending Strong Earthquake by Combining the Seismicity Order Parameter Variability Analysis with Earthquake Networks and Nowcasting: Application in the Eastern Mediterranean. Appl. Sci. 2021, 11, 10093. https://doi.org/10.3390/app112110093

Chouliaras, G.; Skordas, E.S.; Sarlis, N.V. Earthquake Nowcasting: Retrospective Testing in Greece. Entropy 2023, 25, 379. https://doi.org/10.3390/e25020379

which employ the modern method of Earthquake Nowcasting

Rundle, J. B., Turcotte, D. L., Donnellan, A., Grant Ludwig, L., Luginbuhl, M., and Gong, G. (2016), Nowcasting earthquakes, Earth and Space Science, 3, 480– 486, https://doi.org/10.1002/2016EA000185

suggested on the basis of natural time

Varotsos P.A., Sarlis N.V. and Skordas E.S., Natural Time Analysis: The new view of time. Precursory Seismic Electric Signals, Earthquakes and other Complex Time-Series (Springer-Verlag, Berlin Heidelberg) 2011.  https://doi.org/10.1007/978-3-642-16449-1

in order to estimate seismic hazard in Greece and the area around Greece. These should be also mentioned in the list of reference “[1-12]” in line 29 for the readers’ completeness of information.

2)Figures 1 and 2 should be moved (from Section 2) to the second page of the ms in Section 1, just after the second paragraph of the Introduction in which they are mentioned.

3)In the paragraph starting in line 46 and ending in line 77, the acronyms B.C. and A.D. should be replaced by the religious neutral terms B.C.E. (Before Current Era) and C.E. (Current Era), respectively.

4)In Fig. 4, the “HELPOS catalogue norm” plays an important role in understanding the figure. However, these results -probably coming from Ref. [14] are not publicly available since no link is provided for Ref.[14]- are not reproducible by the readers. The authors should fix this data availability issue.

5)Please define the acronym GMPE that appears in line 201 for the first time.

6)The authors’ flow of understanding the ms will be improved if in line 256 just after the acronym REMTH the following text is added “, for their location see Figure 10 that will be displayed later.”

7)In Figures 10 and 11, in the left scale of all panels there are letters missing from the labels 40.5 and 41.5 degrees. Moreover, in the right scale the number 5 and the symbol of degrees is missing in the same labels.

8)Since the audience of GeoHazards is multi-disciplinary, please define epsilon in line 451.

9)The supplementary Table S1 to S12 mentioned in line 466 were not available for review.

10)Please define acronym TR that appears in Figures 16, 17 and Table 3.

11)Please define the exact physical meaning of the parameter R in the caption of Table 3 in line 513 as well as its units.

12)Please define all the new symbols and acronyms that appear in Table 4 in the caption of this Table in line 540.

13)There is no Conclusion section in the ms. Please either rename Section 4 to Discussion and Conclusions in line 549 or add a short Conclusion section after line 597.

14) The text in lines 605 to 607 “Please turn to … the work reported.” should be removed.

In summary, I will be glad to suggest publication of a ms revised along the lines mentioned above.

Minor editing of English language required

Author Response

We would like to thank the reviewer for his/her comments on our manuscript and the time he/she spent on it. Please, in the following find our response. 

1) The references mentioned by the reviewer are related to earthquake forecasting based mainly on geophysical methods. Our work deals with probabilistic seismic hazard assessment which describes how strong and how often a region or site is struck by an strong earthquake and relative strong motion data (e.g. pga, pgv or any other). Thereby, we believed that the proposed references are not relevant to the present manuscript. 

2) We moved figures 1 and 2 in section 1, as proposed by the reviewer.

3) The acronyms have been modified accordingly.

4) The earthquake catalogue of HELPOS comes from ref. 15. A link has been added to the reference list, according to the reviewer's suggestion.

5) The term Ground Motion Prediction Equations (GMPEs) appears for the first time in the main body of the manuscript, along with its acronym, in line 82.

6) The acronym REMTH is explained in line 35 and the region is depicted in Figure 1, in the embedded zoomed map. The caption of figure 1 has been modified so that it is clear the REMTH is presented there.

7) Figures 10 and 11 have been corrected according to the reviewer's comment.

8) Epsilon is defined as "ground motion variability" in line 490 of the revised manuscript.

9) We have submitted the supplement file along with the original and the revised manuscript.

10) Definition of TR is given in line 512 of revised manuscript.

11) The definition of R is given in line 490 of the revised and the original manuscript. Units have been added in the relevant column of Table 3.

12) The symbols have been explained within the text between lines 564 and 572, according to the reviewer's suggestion.

13) According to the template of Geohazards, the Conclusions section is optional, whereas the Discussion is mandatory. Nevertheless, the last section of the manuscript has been renamed.

14) These text lines were removed, according to the reviewer's comment.

Reviewer 4 Report

Summary

This is a well-written research paper with clear and understandable language, clear figures, and tables that represent accurately the results and the study design is appropriate for answering the research questions. Both Summary and Conclusions are clearly written, giving accurate information on the research and the results, without spin for the reviewer and the reader as long as it gets published and gets accessed to a broader audience.

Minor Changes

The issues I‘ve found reading the paper are of minor importance:

1. P9L238-239 “For the present study, the lowest earthquake magnitude M0 which was considered 238 for all the seismic sources was M4.5” I guess you mean Moment Magnitude (Mw).

2. Examine the possibility at the final stage of proofreading to include Figures 11-13 in a single page (1 figure per page instead of 11a,b and c, and d in different ones) with the appropriate resolution. 

Overall, the soundness of the methodology and the conclusions can be supported by the results, in a region surrounded by significant fault zones and high vulnerability.

Therefore, I recommend this research paper be published after some minor changes would be carried out.

Kind regards

The quality of English Language is appropriate for a scientific paper, and some minor editing can be carried out at the final stage of proofreading. 

Author Response

We would like to thank the reviewer for his/her comments. In the following, the response to the reviewer's comments are given:

  1. The type of earthquake magnitude which was considered in our study is the Moment Magnitude (Mw). The symbol M0 is given in conjunction with equation (2) where the integration along the earthquake magnitude is made between the minimum and the maximum considered magnitudes, which are given as M0 and Mmax, respectively.
  2. We move figures 12a and 12b, so that they are shown in the same page. This was also possible for figure 13. However, this was not possible for figure 10 and 11 as it would require to reduce the size of the figures and their visibility.